# Beyond Imitation: Recovering Dense Rewards from Demonstrations

## Abstract

Conventionally, supervised fine-tuning (SFT) is treated as a simple imitation learning process that only trains a policy to imitate expert behavior on demonstration datasets. In this work, we challenge this view by establishing a fundamental equivalence between SFT and Inverse Reinforcement Learning. We prove that the SFT objective is a special case of Inverse Q-Learning, which implies that the SFT process does not just learn a policy, but also an implicit, dense, token-level reward model that explains the expert demonstrations. We then show how to recover this dense reward signal directly from the SFT model by formulating a baseline-relative reward function. The availability of such a dense reward model offers numerous benefits, providing granular credit assignment for each token generated. We demonstrate one key application by using these recovered rewards to further improve the policy with reinforcement learning. Our method, Dense-Path REINFORCE, consistently outperforms the original SFT models on instruction-following benchmarks. This work reframes SFT not merely as policy imitation but as a powerful reward learning mechanism, opening new possibilities for leveraging expert demonstrations.

## 1 Introduction

Large Language Models (LLMs) (Liu et al., 2024; Comanici et al., 2025; Achiam et al., 2023) have rapidly developed from research prototypes to general-purpose assistants that plan, reason, and generate helpful responses across domains. A significant driver of these capabilities is *post-training on demonstrations*—often called *Learning from Demonstrations* (LfD)—where a pretrained model is refined to follow expert responses (Ouyang et al., 2022; Chen et al., 2024). In practice, LfD is implemented almost exclusively as *Supervised Fine-Tuning* (SFT): teacher-forced maximum likelihood on expert tokens conditioned on prompts. Because SFT matches expert sequences, it is commonly framed as *imitation learning* (Xiao et al., 2024; Shaikh et al.; Sun, 2024) in which the model learns only to mimic expert behavior.

This paper argues that the imitation-only view is incomplete. We show that, under standard assumptions for token-level generation, SFT admits a precise interpretation through the lens of *Inverse Reinforcement Learning* (IRL) (Ng & Russell, 2000). Specifically, on the token Markov decision process (MDP) without discount, the token-level SFT objective is *equivalent* to optimizing the reduced objective of Inverse Soft-$Q$ Learning (IQ-Learn) (Garg et al., 2021). In this view, SFT does more than fit a policy: it implicitly learns a *dense token-level reward* that rationalizes expert demonstrations, aligning SFT with the credit-assignment perspective of MaxEnt IRL and GAIL (Ziebart et al., 2008; Ho & Ermon, 2016).

The IQ-Learn perspective also yields a valid recipe for further improving an SFT policy. First, we prove a *dual-contraction* property of the IQ-Learn saddle: the error of the reward estimation is bounded by the policy's occupancy error, so a reasonably accurate policy implies an even more stable reward estimation (near the saddle). Second, we show how to *recover a dense reward* directly from the trained SFT model. Using the soft-optimality identity and potential-based shaping (Ng et al., 1999), the teacher's token log-probability decomposes as the task reward plus a telescoping potential value function. This implies two design choices. (i) We eliminate the value term via shaping, which keeps the token reward dense and avoids tricky value estimation. (ii) To avoid the length bias of raw log-likelihoods (non-positive by construction) and stabilize credit assignment, we use a

*baseline-relative* reward where baseline is a checkpoint during SFT training. This choice measures *incremental performance* gained during SFT and empirically reduces variance. Together, these results justify a simple reinforcement step that stays in the LfD setting: we optimize the SFT policy with *token-level, undiscounted* REINFORCE (Williams, 1992; Ahmadian et al., 2024) using the dense baseline-relative reward.

We evaluate this recipe on four pretrained LLMs and four public instruction-following benchmarks using the same demonstration data for SFT and RL. Despite operating strictly in the LfD setting, the resulting policy improves over the SFT model in head-to-head win rate and standardized multi-turn scores, showing competitiveness with other LfD baselines such as SPIN (Chen et al., 2024) and GSIL (Xiao et al., 2024).

Our primary contributions are as follows: (i) We establish formal equivalence between token-level SFT and the reduced objective of IQ-Learn on the token MDP, reframing SFT as implicit dense reward learning rather than pure imitation. (ii) We prove that near the IRL saddle, the reward estimation error is bounded by the policy occupancy error, explaining why rewards recovered from an SFT policy can be more stable than the policy itself. (iii) We construct meaningful token-level rewards through reward shaping theory and the strategic selection of a reward baseline. (iv) We instantiate these insights in a minimal reinforcement learning algorithm that uses token-level, undiscounted baseline-relative reward as the learning objective. (v) Across four pretrained backbones and four instruction-following evaluations, this method consistently improves over SFT and matches or exceeds other LfD baselines.

## 2 RELATED WORK

**Independent and contemporaneous (posted earlier) work: reward signals inside LLMs.** Before this work, Li et al. (2025) also argues that LLMs contain useful token-level reward signals through the lens of IRL. While we share the same theoretical framework on the equivalence of IRL and SFT, the theoretical part of our work was developed *independently* before the publication of their work. Their experimental focus is to extract sentence-level rewards through aggregation of token-level rewards, often from instruction-tuned LLMs, and then use the aggregated reward for policy improvement and reward-based pair-wise classification. Our setting and emphasis are different: we operate in LfD with pretrained backbones and develop a shaping- and baseline-based reward construction that makes dense rewards workable for policy improvement in practice.

**Imitation learning and LfD for LLMs.** Beyond direct cloning, several LfD approaches leverage self-generated data to improve a policy without requiring explicit preference pairs. These methods reframe the learning problem to go beyond the simple negative log-likelihood objective of SFT: SPIN uses self-play fine-tuning to convert weaker models into stronger ones (Chen et al., 2024). (Li et al., 2024) found that SPIN is a special case of IRL; however, they still focus on the gap between policy and expert at the sample level. GSIL also uses both real demonstration data and self-generated model data, but formulates the problem from an imitation learning perspective (Xiao et al., 2024). Our work differs in both analysis and mechanism: we remain strictly in the LfD setting, but *re-interpret* SFT through an IRL lens (SFT ≡ IQ-Learn on the token MDP).

**Preference-based post-training (RLHF, DPO family, GRPO).** Another line of work treats post-training as optimization from *pairwise* human (or AI) preferences. PPO-based RLHF (Ouyang et al., 2022) fits a reward model and then optimizes the policy with reinforcement learning. DPO (Rafailov et al., 2023) replaces explicit reward learning and online rollouts with a direct, classification-style objective. Recent GRPO-style methods explore preference optimization without an explicit critic: *group relative* policy optimization has been used in scaling efforts to stabilize on-policy updates via group-normalized advantages (Shao et al., 2024). These methods require preference data or verifiable rewards and thus are outside our scope.

**Connection of reinforcement learning and SFT.** Xiao et al. establishes a theoretical connection between reinforcement learning and imitation learning, revealing that RLHF implicitly performs imitation learning on the preference data distribution. Qin & Springenberg (2025) unifies SFT with RL through importance sampling. These studies are somewhat related to our work, but they primarily focus on the relationship between RL and SFT, whereas we analyze SFT from the perspective of IRL.

## 3  PRELIMINARIES

This section introduces the minimal background needed to follow our methodology and proofs. We formalize the token-level MDP for autoregressive generation, recall the entropy-regularized optimality equations, restate MaxEnt IRL in an occupancy form, explain the $Q$-space reduction used by IQ-Learn.

**Problem setup and notation.**    We model generation as a finite-horizon token MDP $(\mathcal{S}, \mathcal{A}, f, \rho_0)$ with deterministic concatenation $f(s, a) = s|a$ and horizon $H$. A state $s_t$ is the prompt plus the tokens generated so far, the action $a_t$ is the next token, and an LLM induces a policy $\pi(a \mid s)$. We write the (state–action) *occupancy measure* of policy $\pi$ as

$$\rho_\pi(s, a) \;=\; \sum_{t=0}^{H-1} \Pr_\pi(s_t = s, a_t = a), \qquad \langle \rho_\pi, r \rangle \;:=\; \sum_{s,a} \rho_\pi(s, a)\, r(s, a).$$

For any real-valued function $Q : \mathcal{S} \times \mathcal{A} \to \mathbb{R}$, define the soft value $V(s) = \log \sum_a \exp Q(s, a)$ and the Boltzmann policy $\pi_Q(a \mid s) \propto \exp Q(s, a)$ (temperature fixed to 1 throughout).

**Soft-optimality equations.**    In entropy-regularized control (Haarnoja et al., 2017), optimizing $\mathbb{E}_{a \sim \pi(\cdot|s)}[Q^\star(s, a)] - \beta \, \mathrm{H}(\pi(\cdot \mid s))$ over $\pi(\cdot \mid s)$ yields the familiar logit form of the optimal policy and value:

$$\pi^\star(a \mid s) = \exp\!\Big( \tfrac{1}{\beta}\big(Q^\star(s, a) - V^\star(s)\big) \Big), \qquad V^\star(s) = \beta \log \sum_{a \in \mathcal{A}} \exp\!\big( \tfrac{1}{\beta} Q^\star(s, a) \big). \tag{1}$$

That is, $\pi^\star(\cdot \mid s)$ is the Boltzmann distribution over $Q^\star(s, \cdot)$ and $V^\star(s)$ is the corresponding log-partition. A full derivation is provided in Appendix A.2.

**MaxEnt IRL in occupancy space.**    Maximum-entropy IRL seeks a reward $r$ that rationalizes expert behavior by comparing expert and learner occupancies while keeping the policy stochastic via entropy Ziebart et al. (2008); Ho & Ermon (2016):

$$L(\pi, r) \;=\; \langle \rho_E - \rho_\pi, \; r \rangle \;-\; H(\pi) \;-\; \psi(r). \tag{2}$$

Here $\psi$ is a convex regularizer on rewards (for identifiability/stability). The saddle point of (2) matches occupancies ($\rho_{\pi^\star} = \rho_E$) and produces a reward $r^\star$ unique up to potential-based shaping.

**IQ-Learn: a $Q$-space reduction.**    IQ-Learn re-parameterizes the IRL objective so that, after minimizing over $\pi$, one optimizes a *concave* functional of $Q$ Garg et al. (2021). The policy minimizer is $\pi_Q(a \mid s) = \exp(Q(s, a) - V(s))$, and the reduced objective $J^*(Q)$ aggregates the "soft-advantage" $Q(s, a) - V(f(s, a))$ along expert trajectories. On a deterministic token tree ($f(s, a) = s'$), telescoping arguments become particularly simple and will later allow us to show that *token-level SFT is equivalent to* maximizing $J^*(Q)$ under a linear conjugate (Step 1).

## 4  METHODOLOGY

**High-level outline.** Our methodology follows three steps. **(S1)** We show that the token-level SFT objective is *equivalent to* the reduced IQ-Learn objective under a mild regularizer. **(S2)** Within the IRL/IQL framework, we prove that the *reward estimation error* is controlled by the *policy error* in occupancy space. **(S3)** We extract a baseline-relative, log-likelihood based dense reward (Chan et al., 2024) from the SFT model and show that any improvement on this proxy transfers to improvement under the true objective.

### 4.1  STEP 1: SFT IS EQUIVALENT TO A SPECIAL CASE OF IQ-LEARN

**Statement.** Let $J^*(Q)$ denote the reduced IQ-Learn objective after minimizing over $\pi$ (Garg et al., 2021). On the token MDP with $\gamma = 1$ and a linear conjugate (i.e., no extra reward regularization beyond convexity), maximizing $J^*(Q)$ is *equivalent to* maximizing the teacher-forced log-likelihood on expert tokens:

$$\max_Q J^*(Q) \;\equiv\; \max_Q \; \mathbb{E}_{(s,a) \sim \rho_E}\big[ \log \pi_Q(a \mid s) \big],$$

where $\pi_Q(a \mid s) \propto \exp Q(s, a)$ and $V(s) = \log \sum_a e^{Q(s,a)}$.

**Intuition.** The reduction $J^*(Q)$ aggregates a "soft-advantage" term of the form $Q(s, a) - V(f(s, a))$ along expert trajectories. On a deterministic token sequence, the value contributions telescope across time, and the identity $\log \pi_Q(a \mid s) = Q(s, a) - V(s)$ converts the objective into the SFT log-likelihood.

**Proposition 1** (SFT ≡ IQ-Learn with a linear conjugate). *On the token MDP with discount rate $\gamma = 1$, maximizing $J^*(Q)$ is* equivalent to *minimizing the token-level SFT loss* $\mathcal{L}_{\mathrm{SFT}}(\theta) = \mathbb{E}_{(s,a)\sim\rho_E}[-\log\pi_\theta(a \mid s)]$, where $\pi_\theta(a \mid s) \propto \exp Q_\theta(s, a)$.

*Proof.* See Appendix A.4 for a complete derivation via telescoping and the identity $\log \pi_Q = Q - V$. 

**Takeaway.** SFT is not only policy imitation: it is *exactly* the $Q$-space objective of an IQ-Learn instance on the token MDP. Consequently, SFT logits can be treated as a $Q$-function without leaving the IRL/IQL lens, consistent with the token-level perspective in *From r to $Q^*$* (Rafailov et al.).

## 4.2 STEP 2: REWARD ERROR IS CONTROLLED BY POLICY ERROR (IRL VIEW)

We adopt the convex-analytic IRL objective (Ho & Ermon, 2016):

$$L(\pi, r) = \langle \rho_E - \rho_\pi, r \rangle - H(\pi) - \psi(r). \tag{3}$$

Let $r^\star$ be a reward at the IRL saddle. For any $\pi$, let the reward best response be $\widehat{r}(\pi) := \arg\max_r L(\pi, r)$. Measure the *policy error* by $\varepsilon_\pi := \|\rho_\pi - \rho_E\|_*$ and the *reward error* by $\varepsilon_r := \|\widehat{r}(\pi) - r^\star\|$, where $\|\cdot\|$ and $\|\cdot\|_*$ are dual norms.

**Theorem 2** (Dual contraction: reward error ≤ policy error). *If $\psi$ is $\mu$-strongly convex in $\|\cdot\|$, then for any policy $\pi$,*

$$\left\|\widehat{r}(\pi) - r^\star\right\| \leq \frac{1}{\mu}\left\|\rho_\pi - \rho_E\right\|_*.$$

*Proof.* By first-order optimality for the reward player, $\nabla\psi(\widehat{r}(\pi)) = \rho_E - \rho_\pi$ and $\nabla\psi(r^\star) = \rho_E - \rho_{\pi^\star}$. At the saddle $\rho_{\pi^\star} = \rho_E$, so $\nabla\psi(r^\star) = 0$ and hence $\nabla\psi(\widehat{r}(\pi)) - \nabla\psi(r^\star) = \rho_E - \rho_\pi$. Strong convexity implies $\mu$-strong monotonicity of $\nabla\psi$; applying Hölder's inequality in dual norms yields the claim. See Appendix A.6 for details. 

**Takeaway.** Learning a reward is at least as stable as learning the policy near the saddle—precisely the property we need before using the (SFT-derived) reward to further improve the policy.

## 4.3 STEP 3: FROM AN SFT-DERIVED DENSE REWARD TO POLICY IMPROVEMENT

**(A) Using SFT logits as a reward via potential shaping.** Combining the soft Bellman identity with $\log\pi_{\mathrm{SFT}}(a \mid s) = Q_{\mathrm{SFT}}(s, a) - V_{\mathrm{SFT}}(s)$ yields

$$\log\pi_{\mathrm{SFT}}(a_t \mid s_t) = r(s_t, a_t) + \big(V_{\mathrm{SFT}}(s_{t+1}) - V_{\mathrm{SFT}}(s_t)\big), \tag{4}$$

so $\log\pi_{\mathrm{SFT}}$ is a shaped version of the task reward and shares the same optimal policies (Ng et al., 1999). This lets us use SFT logits as dense token rewards without explicitly estimating values.

**(B) Why we eliminate $V$ and choose REINFORCE.** For $\gamma = 1$, Step 1 guarantees the SFT↔IQ-Learn equivalence; however, Monte-Carlo returns for early tokens are larger in magnitude than for later tokens:

$$\sum_{k=t}^{H-1} \log\pi_{\mathrm{SFT}}(a_k \mid s_k) = \sum_{k=t}^{H-1} r(s_k, a_k) - V_{\mathrm{SFT}}(s_t) \quad (V_{\mathrm{SFT}}(s_H) = 0),$$

so returns differ by a state-dependent constant $-V_{\mathrm{SFT}}(s_t)$. Fitting a critic (as in PPO) to such heteroskedastic targets is difficult, especially if $V_{\mathrm{SFT}}$ is noisy. Using REINFORCE avoids a critic entirely; Appendix A.8 shows that the policy gradient with reward $\log\pi_{\mathrm{SFT}}$ equals that with reward $r$ up to a baseline $b_t(s_t) = V_{\mathrm{SFT}}(s_t)$.

**(C) A baseline-relative dense reward.** Directly maximizing $\sum_t \log\pi_{\mathrm{SFT}}(a_t \mid s_t)$ favors short sequences (token log-probabilities are non-positive). We therefore use

$$\widehat{r}(s, a) = \log\pi_{\mathrm{SFT}}(a \mid s) - \log\pi_{\mathrm{ref}}(a \mid s), \tag{5}$$

Response: Eliza's regular rate is $10 per hour. She worked for 40 hours at this rate, so she earned 40 x 10 = 400 dollars. For the remaining 5 hours, she gets 1.2 times her regular rate. So she earns 5 x 1.2 x 10 = 60 dollars for overtime. Total earnings are 400 + 60 = 460 dollars. The answer is 460.

Response: Eliza's regular rate is $10 per hour. She worked for 40 hours at this rate, so she earned 40 x 10 = 400 dollars. For the remaining 5 hours, she gets 1.2 times her regular rate. So she earns 5 x 2.2 x 10 = 110 dollars for overtime. Total earnings are 400 + 110 = 510 dollars. The answer is 510.

Figure 1: Credit assignment in Dense-Path REINFORCE (Best viewed in color). We provide two answers to a math question. The left is the correct response, and on the right is our modified response. Each token is colored according to the baseline-relative dense reward as expressed in Eq. (5) (darker red means higher reward), using the trained SFT model and SFT checkpoint. We see that the model correctly identifies the erroneous number, without much change to the reward value of the other tokens, which indicates the ability to do credit assignment.

where $\pi_{\text{ref}}$ is a SFT checkpoint with half training samples. This cancels length bias, measures incremental competence, and empirically reduces variance. Appendix A.9 bounds the return shift by $\|V_{\text{SFT}} - V_{\text{ref}}\|_\infty$.

**Illustrative example.** We provide two visualizations in Figure 1 to intuitively demonstrate how $\widehat{r}$ performs credit assignment at the token level. The reward is calculated by SFT-trained LLaMA-3.1-8B and its checkpoint as the baseline. The original question is: "Eliza's rate per hour for the first 40 hours she works each week is $10. She also receives overtime pay at 1.2 times her regular hourly rate. If Eliza worked for 45 hours this week, how much are her earnings for this week?" The left side shows the correct answer, while the right displays our modified incorrect answer. When calculating overtime pay, the incorrect answer erroneously added 1.2 times to the original amount, leading to an incorrect result. Analysis reveals that multiplying 5 by 2.2 resulted in a low reward assigned to the integer part "2", indicating the proposed reward can identify this as an erroneous step. Furthermore, although the subsequent calculations in the incorrect answer are correct, the final result remains wrong, so the assigned reward is lower than that for the correct answer. Additionally, we observe that the "5" in the third row receives a relatively high reward. This "5" does not actually appear in the original question; it skips a calculation step ("45-40") to derive overtime hours. Nevertheless, $\widehat{r}$ still accurately identifies this as a valid step.

**(D) Safe improvement: transferring proxy gains to true gains.** Let $\pi'$ be an update that increases the proxy return by $\Delta_{\widehat{r}} := J_{\widehat{r}}(\pi') - J_{\widehat{r}}(\pi) \geq m$. The performance-difference identity in occupancy space gives

$$J_r(\pi') - J_r(\pi) \ \geq \ m \ - \ 2H \left\| r - \widehat{r} \right\|_\infty, \tag{6}$$

since $\|\rho_{\pi'} - \rho_\pi\|_1 \leq 2H$ for a length-$H$ token MDP. See Appendix A.7 for a complete proof.

**Takeaway.** (1) $\log \pi_{\text{SFT}}$ is a shaped version of the task reward, so it is a valid dense token reward; (2) An SFT checkpoint baseline stabilizes learning and removes the EOS pathology; (3) any optimizer that increases the proxy return (REINFORCE in our case) *safely* improves the true objective once the proxy is accurate enough.

## 5 EXPERIMENTS

### 5.1 EXPERIMENTAL SETUP

**Data.** We adopt **Open-Orca** and subsample **100k** (prompt, demonstration) pairs for SFT and for the RL rollouts (same prompts; no new prompts are introduced in RL). Open-Orca is a large-scale open dataset derived from FLAN-style sources augmented with synthetic expert demonstration from LLMs (Mukherjee et al., 2023). Using the same pool of prompts ensures the effect of our dense, baseline-relative reward does not come from newly introduced prompts.

**Backbones (pretrained only).** To ensure that learning signals from SFT-style demonstrations remain informative, we evaluate only on *foundation (pretrain)* checkpoints (not instruction-tuned). Concretely, we use four sizes/families representative of current open models: `LLaMA-3.1-8B` (Dubey

---

**Algorithm 1** Dense-Path REINFORCE

---

**Require:** Expert dataset $\mathcal{D}_E$, base model $\theta_{\text{base}}$, total SFT steps $N$, horizon $H$, baseline fraction $\alpha \in (0, 1)$ (default 0.5), discount $\gamma \leftarrow 1$, KL weight $\lambda_{\text{KL}} \geq 0$

**Ensure:** Fine-tuned policy $\pi_\phi$

1: **SFT stage:** Fine-tune $\theta_{\text{base}}$ on $\mathcal{D}_E$ for $N$ steps; set teacher $\pi_{\text{SFT}} \leftarrow \pi_{\theta_N}$. Save the checkpoint with half training steps as reference $\pi_{\text{ref}} \leftarrow \pi_{\theta_{\lfloor \alpha N \rfloor}}$.

2: **Initialize actor:** $\pi_\phi \leftarrow \pi_{\text{SFT}}$; freeze $\pi_{\text{SFT}}$ and $\pi_{\text{ref}}$.

3: **for** training iteration $= 1, 2, \ldots$ **do**

4:     Sample a batch of prompts $\{x_i\}_{i=1}^B$; roll out trajectories $\tau_i = (s_0, a_0, \ldots, s_{T_i-1}, a_{T_i-1})$ using $\pi_\phi$.

5:     **for all** tokens $(s_t, a_t)$ in each $\tau_i$ **do**

6:         **Baseline-relative token reward (Eq. (5)):** $\widehat{r}_t \leftarrow \log \pi_{\text{SFT}}(a_t \mid s_t) \; - \; \log \pi_{\text{ref}}(a_t \mid s_t)$

7:     **end for**

8:     **Per-token returns:** For each trajectory $i$, compute $G_t \leftarrow \sum_{k=t}^{T_i-1} \widehat{r}_k$ for all $t$.

9:     **Total objective (token-level):**

$$\mathcal{L}(\phi) \; = \; -\frac{1}{B} \sum_{i=1}^B \sum_{t=0}^{T_i-1} \log \pi_\phi(a_t \mid s_t) \, G_t$$

10:     **Gradient step:** Update $\phi$ by Adam on $\nabla_\phi \mathcal{L}(\phi)$.

11: **end for**

---

et al., 2024), `Qwen-2.5-7B` (Yang et al., 2024), `Mistral-7B-v0.1` (Jiang et al., 2024), and `Gemma-3-4B` (Team et al., 2025).

**Baselines.** We compare with: (i) **SFT** (teacher-forced cross-entropy on the 100k set); (ii) **SPIN** (self-play fine-tuning from demonstrations) (Chen et al., 2024); (iii) **GSIL** (self-imitation learning on demonstrations) (Xiao et al., 2024); and (iv) **SR** (sentence-level REINFORCE): it uses the same baseline-relative reward as our method but assigns the *entire trajectory return only at EOS*, i.e., a sparse reward delivered once per sequence (conceptually close to PPO-style sparse credit assignment). We also test the performance of PPO using sentence-level baseline-relative reward as reward signals, but it doesn't show significant differences with REINFORCE. All baselines use the same prompts and demonstrations.

**Our method.** We implement the REINFORCE variant described in §4.3 with token-level returns (undiscounted, $\gamma = 1$), and *baseline-relative dense rewards* $\widehat{r}(s, a) = \log \pi_{\text{SFT}}(a \mid s) - \log \pi_{\text{ref}}(a \mid s)$ (SFT checkpoint as $\pi_{\text{ref}}$). We employ a modern RLHF stack based on **Open-RLHF**'s `REINFORCE++` implementation (KL regularization, clipping, and standard stability tricks) (Hu et al., 2024; 2025).

**Evaluation.** We use four public instruction-following evaluations: **AlpacaEval**(Li et al., 2023), **Arena-Hard** (Li et al.), **LIMA** prompts (Zhou et al., 2023), and **MT-Bench** (standardized 1–10 scoring) (Zheng et al., 2023). For *AlpacaEval*, *Arena-Hard*, and *LIMA*, we report *pairwise win rate versus the SFT model* using **GPT-4o** as the judge (temperature 0; ties count as 0.5) (Achiam et al., 2023). For *MT-Bench*, we report the standard 1–10 score using the official scripts. Following the general test setting for instruction following tasks, decoding uses a temperature 0.7 with a fixed max generation length. To minimize tuning bias, **all backbones share the same hyperparameters** (Appendix Table 4); this avoids per-model over-tuning.

### 5.2 MAIN RESULTS

**Detailed analysis of Table 1.** **(i) LfD gains across backbones.** Across all four *pretrained* backbones, our token-level method (DPR) improves over the SFT policy on the three win-rate benchmarks and MT-Bench scores, confirming that *dense, baseline-relative* rewards extracted from SFT logits can further upgrade the policy without introducing new prompts. Typical gains over SFT range from single digits on easier benchmarks to double digits on harder benchmarks (e.g., *Arena-Hard*).

Table 1: **Instruction following results across four pretrained backbones.** For **AlpacaEval**, **Arena-Hard**, and **LIMA**, we report *GPT-4o win rate (%)* versus the SFT model. For **MT-Bench**, we report the standard *1–10 score*. All methods train on the same 100k Open-Orca samples. Bold = best, underline = second best, per model group.

| Method | AlpacaEval | Arena-Hard | LIMA | MT-Bench |
|---|---|---|---|---|
| | GPT-4o Win Rate (%) ↑ | | | Score (1–10) ↑ |
| *LLaMA-3.1-8B* | | | | |
| SFT | - | - | - | 5.74 |
| SPIN | 55.2 | 53.3 | 53.0 | 5.81 |
| GSIL | 58.1 | 56.7 | 61.0 | 5.92 |
| SR | 57.9 | 60.3 | 60.8 | 5.96 |
| DPR | **60.6** | **62.5** | **62.7** | **6.01** |
| *Qwen-2.5-7B* | | | | |
| SFT | - | - | - | 6.83 |
| SPIN | 55.5 | 51.4 | 57.5 | 6.98 |
| GSIL | 56.2 | 53.3 | 56.2 | 7.01 |
| SR | 55.9 | 54.6 | 54.0 | 7.09 |
| DPR | **57.3** | **55.2** | **59.8** | **7.29** |
| *Mistral-v0.1-7B* | | | | |
| SFT | - | - | - | 5.23 |
| SPIN | 58.3 | 55.0 | 53.0 | 5.45 |
| GSIL | 59.2 | 54.8 | 54.0 | 5.43 |
| SR | 46.6 | 49.8 | 47.3 | 5.14 |
| DPR | **61.0** | **60.7** | **59.3** | **5.65** |
| *Gemma-3-4B* | | | | |
| SFT | - | - | - | 5.32 |
| SPIN | 58.6 | 54.7 | 58.7 | 5.47 |
| GSIL | 60.3 | 57.1 | 60.8 | **5.56** |
| SR | 65.6 | 58.0 | 64.5 | 5.48 |
| DPR | **66.7** | **58.9** | **66.8** | 5.54 |

**(ii) Dense vs. sparse credit assignment.** Relative to *SR* (EOS-only return), DPR achieves systematically higher win rates and MT-Bench scores, supporting the hypothesis that *token-level* returns (with $\gamma=1$) offer better credit assignment than sparse, trajectory-level returns. Notably on *Mistral-v0.1-7B*, DPR has a large gap vs. SR on four benchmarks, indicating that per-token shaping is especially beneficial when the base model underfits demonstrations.

**(iii) LfD baselines (SPIN/GSIL).** Compared with *SPIN* and *GSIL*, both LfD methods that also use only demonstrations, DPR is competitive or superior on most benchmarks. The advantage is most pronounced on *Arena-Hard*, which is known to better separate models and correlate with Arena human preferences. This suggests that our reward extraction provides a stronger, more stable learning signal than self-play or self-imitation on the same prompt/demonstration pool.

**(iv) MT-Bench improvements are consistent though modest.** On MT-Bench (1–10), DPR shows small but consistent absolute gains over SFT across backbones (typically +0.2 to +0.5), in line with the expectation that general multi-turn quality improves when local token decisions are better rewarded.

## 5.3 ABLATION STUDY

**Findings.** **(a) Effect of eliminating $V$.** Compared to *w/DPR*, *w/V* drops on all backbones and metrics (typically by 2–7 win-rate points), corroborating our theory that the potential term $V$ induces position-dependent return shifts that are hard to fit and unnecessary under $\gamma=1$ (cf. §4.3 and Appendix A.8). **(b) Necessity of the baseline.** Removing the SFT checkpoint baseline (*wo/Baseline*) causes large drops (often 10–15 win-rate points). This matches the EOS pathology: because token

Table 2: **Ablations on reward shaping and baseline.** *w/DPR*: our full method. *w/V*: do *not* eliminate the potential term $V$ (i.e., optimize with raw reward $r(s_t, a_t) = \log \pi_{\text{SFT}}(a_t \mid s_t) + \left(V_{\text{SFT}}(s_t) - V_{\text{SFT}}(s_{t+1})\right)$), without using shaping to cancel $V_{\text{SFT}}(s_t) - V_{\text{SFT}}(s_{t+1})$. *wo/Baseline*: remove the halfway SFT baseline (use only $\log \pi_{\text{SFT}}$ as reward). Across backbones and benchmarks, *w/V* consistently underperforms *w/DPR*, indicating that $V$ is noisy and its position-dependent returns harm stability; *wo/Baseline* degrades substantially, consistent with the EOS pathology and length bias discussed in §4.3.

| Variant | AlpacaEval ↑ | Arena-Hard ↑ | LIMA ↑ | MT-Bench ↑ |
|---|---|---|---|---|
| *LLaMA-3.1-8B* | | | | |
| w/DPR | 60.6 | 62.5 | 62.7 | 6.01 |
| w/V | 58.8 | 59.3 | 59.7 | 5.83 |
| wo/Baseline | 49.8 | 46.4 | 46.0 | 5.67 |
| *Qwen-2.5-7B* | | | | |
| w/DPR | 57.3 | 55.2 | 59.8 | 7.29 |
| w/V | 55.0 | 52.9 | 58.0 | 7.12 |
| wo/Baseline | 46.6 | 44.9 | 45.7 | 6.59 |
| *Mistral-7B-v0.1* | | | | |
| w/DPR | 61.0 | 60.7 | 59.3 | 5.65 |
| w/V | 53.9 | 51.8 | 52.3 | 5.47 |
| wo/Baseline | 44.5 | 40.3 | 42.7 | 5.14 |
| *Gemma-3-4B* | | | | |
| w/DPR | 66.7 | 58.9 | 66.8 | 5.54 |
| w/V | 63.5 | 56.0 | 62.2 | 5.51 |
| wo/Baseline | 50.6 | 48.1 | 48.8 | 5.26 |

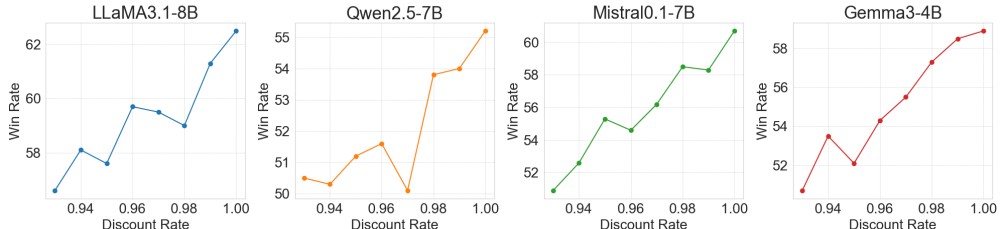

Figure 2: **The effect of reward discount-rate ($\gamma \in \{0.93, 0.94, ..., 1.00\}$) across four backbones.** Performance (win rate vs. SFT, higher is better) peaks at the *undiscounted* setting $\gamma = 1.0$. This is consistent with our analysis: (i) the SFT↔IQ-Learn equivalence is derived for $\gamma = 1$; (ii) with discounting, early tokens are over-rewarded relative to later ones, weakening token-level credit assignment.

log-probs are non-positive, shorter sequences spuriously obtain larger undiscounted returns without the baseline correction; the baseline cancels this length bias and stabilizes updates.

### 5.4 SENSITIVITY ANALYSES

**The effect of reward discount rate.** Undiscounted returns preserve the telescoping structure that underpins our shaping equivalence and avoid compressing late-token contributions. Empirically, as shown in Figure 2, moving from $\gamma < 1$ to 1.0 improves the win rate consistently across models, with larger gains for weaker backbones (e.g., Mistral-7B-v0.1) where late-token guidance matters more.

**The effect of baseline checkpoint selection.** As shown in Figure 3, across backbones, the performance curve is roughly unimodal with a maximum near the checkpoint with around half of the total training samples. This supports the interpretation of our reward as "incremental competence" gained during SFT: too early, the baseline is not competitive enough; too late, the gap collapses and the proxy reward diminishes.

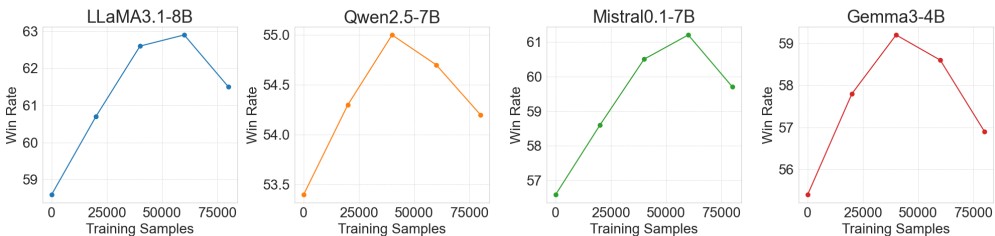

Figure 3: **Baseline checkpoint selection.** We vary the baseline $\pi_{\mathrm{ref}}$ along the SFT training trajectory (x-axis: SFT progress), keeping all else fixed. A baseline trained with around half of the total training samples yields the best results. Intuitively, an *early* baseline is too weak, over-inflating rewards and increasing variance; a *late* baseline is too close to the full SFT, shrinking $\log \pi_{\mathrm{SFT}} - \log \pi_{\mathrm{ref}}$ and reducing signal-to-noise. The midpoint balances *magnitude* and *discriminativeness*, consistent with our bound in Appendix A.9.

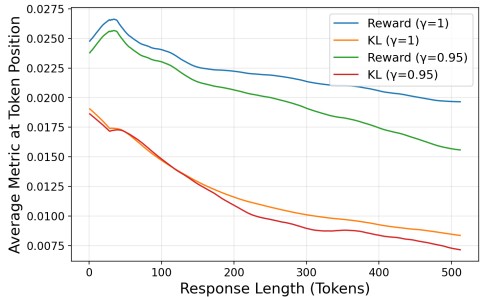

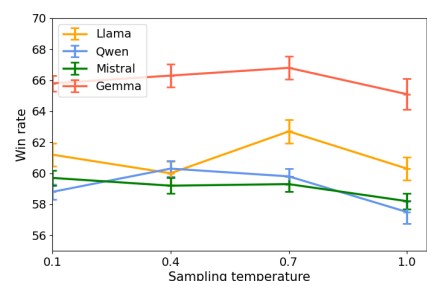

(a) Visualization of the average KL divergence and reward of responses after DPR training.

(b) We vary the generation temperature of both DPR and the SFT baseline when evaluated on the LIMA benchmark.

**The effect of evaluation temperature.** As depicted in Figure 4b, taking the LIMA benchmark as an example, our algorithm demonstrates significant improvements over SFT across different sampling temperatures during evaluation, indicating its robustness to sampling temperature variations. Furthermore, we observe that although the win rate slightly decreases when the sampling temperature is set to 1, it remains markedly superior to the SFT model. This suggests that our model not only enhances sampling efficiency in high-confidence regions but also achieves notable improvements in other areas.

**Analysis of KL divergence and reward with respect to response length.** Previous studies have found that the majority of the contribution from post-training algorithms might be concentrated in the initial response tokens (Qi et al.). As the response length increases, the contribution of these algorithms may begin to diminish. Correspondingly, in our algorithm, this may be related to the discount rate, as a larger discount rate might exacerbate this phenomenon. To substantiate this observation, we compared the response rewards and KL divergence as a function of length when the discount rate was set to 1 and 0.95. As shown in the Figure 4a, the KL divergence decreases rapidly with increasing length. When the discount rate is 1, the model still retains a high reward within a limited KL budget. However, when the discount rate is 0.95, the model exhibits a more pronounced decline in reward. The results indicate that the phenomenon of rewards decreasing with length does indeed exist, but rewards without discounts can mitigate it to some extent.

## 6 CONCLUSION

This paper revisits LfD for LLMs through the lens of IRL. We show that the token-level SFT objective is *equivalent* to the reduced objective of IQ-Learning. In this view, SFT not only fits a policy but also encodes a dense token-level reward signal in its logits. Building on this equivalence, we propose DPR, a REINFORCE variant that uses dense baseline-relative rewards from the SFT model. Empirically, across four pretrained backbones and four public instruction-following benchmarks, DPR consistently surpasses the SFT baseline and is competitive with other LfD methods.

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

# Appendix

## A  FULL PROOFS AND TECHNICAL DETAILS

### A.1  NOTATION, BASIC ASSUMPTIONS, AND IDENTITIES

We work on the finite-horizon token MDP $(\mathcal{S}, \mathcal{A}, f, \rho_0)$ with deterministic transition $f(s, a) = s|a$ and horizon $H$. A trajectory is $\tau = (s_0, a_0, \ldots, s_H)$ with $s_{t+1} = f(s_t, a_t)$ and $s_H$ terminal (EOS or max length). For any policy $\pi$, the *occupancy measure* is

$$\rho_\pi(s, a) = \sum_{t=0}^{H-1} \Pr_\pi(s_t = s, a_t = a), \qquad \langle \rho_\pi, r \rangle = \sum_{s,a} \rho_\pi(s, a)\, r(s, a).$$

For a function $Q : \mathcal{S} \times \mathcal{A} \to \mathbb{R}$, define the log-partition (soft value) and Boltzmann policy

$$V(s) = \beta \log \sum_{a \in \mathcal{A}} e^{Q(s,a)/\beta}, \qquad \pi_Q(a \mid s) = \exp\Big(\tfrac{1}{\beta}\big(Q(s,a) - V(s)\big)\Big),$$

with fixed temperature $\beta > 0$ (we use $\beta = 1$ when not stated). We frequently use the identity

$$\log \pi_Q(a \mid s) = \tfrac{1}{\beta}\big(Q(s,a) - V(s)\big). \tag{7}$$

### A.2  DERIVATION OF EQ. (1) (OPTIMAL SOFT POLICY AND VALUE)

**Setup.**  Fix a state $s$. Consider the convex optimization problem

$$\max_{\pi(\cdot|s) \in \Delta(\mathcal{A})} \sum_a \pi(a \mid s)\, Q^\star(s, a) - \beta \sum_a \pi(a \mid s) \log \pi(a \mid s),$$

subject to (i) $\sum_a \pi(a \mid s) = 1$, (ii) $\pi(a \mid s) \geq 0$ for all $a$. The objective is strictly concave in $\pi(\cdot \mid s)$ because the negative entropy $-\sum \pi \log \pi$ is strictly convex and we *maximize* its negation; hence the maximizer is unique.

**KKT conditions.**  Form the Lagrangian

$$\mathcal{L}(\pi, \lambda, \{\nu_a\}) = \sum_a \pi(a \mid s)\, Q^\star(s, a) - \beta \sum_a \pi(a \mid s) \log \pi(a \mid s) + \lambda\Big(\sum_a \pi(a \mid s) - 1\Big) + \sum_a \nu_a\, \pi(a \mid s),$$

with multipliers $\lambda \in \mathbb{R}$ for the simplex constraint and $\nu_a \geq 0$ for non-negativity. Stationarity for every $a$ gives

$$\frac{\partial \mathcal{L}}{\partial \pi(a \mid s)} = Q^\star(s, a) - \beta\big(1 + \log \pi(a \mid s)\big) + \lambda + \nu_a = 0.$$

*Complementary slackness:* if $\pi^\star(a \mid s) > 0$, then $\nu_a = 0$. Since the optimum has full support under finite $\beta > 0$ (the entropy term forces interior optimum), we set $\nu_a = 0$ for all $a$ and obtain

$$\log \pi^\star(a \mid s) = \tfrac{1}{\beta}\big(Q^\star(s, a) + \lambda - \beta\big).$$

Exponentiating and normalizing by the constraint yields

$$\pi^\star(a \mid s) = \frac{\exp(Q^\star(s, a)/\beta)}{\sum_{a'} \exp(Q^\star(s, a')/\beta)}.$$

Defining $V^\star(s) := \beta \log \sum_{a'} \exp(Q^\star(s, a')/\beta)$ gives the stated softmax policy and the value expression in Eq. (1). This completes the derivation.

### A.3  FROM MAXENT IRL TO THE IQ-LEARN REDUCED OBJECTIVE $J^*(Q)$

In this section, we give a minimal proof modified from IQ-Learn (Garg et al., 2021). We recall the MaxEnt IRL saddle objective

$$L(\pi, r) = \langle \rho_E - \rho_\pi,\ r \rangle - H(\pi) - \psi(r), \tag{8}$$

with a convex reward regularizer $\psi$ for identifiability/stability (Ziebart et al., 2008; Ho & Ermon, 2016). For a fixed $Q$, minimizing $L$ over $\pi$ with the soft entropy yields the Boltzmann policy $\pi_Q$ in (7); the corresponding *reduced* objective over $Q$ (IQ-Learn) is

$$J^*(Q) \;=\; \mathbb{E}_{(s,a)\sim\rho_E}\big[Q(s,a) - V(f(s,a))\big] \;-\; \mathbb{E}_{s_0\sim\rho_0}\big[V(s_0)\big], \tag{9}$$

where $V$ is the log-partition induced by $Q$ and $f$ is the deterministic environment transition. For completeness, we expand all steps below.

**Detailed derivation.** Write the inner minimization over $\pi$ at each state $s$:

$$\min_{\pi(\cdot|s)\in\Delta} \Big\{ -\sum_a \pi(a \mid s)\, Q(s,a) + \beta \sum_a \pi(a \mid s)\log\pi(a \mid s) \Big\} = -\max_{\pi(\cdot|s)}\Big\{ \sum_a \pi(a \mid s) Q(s,a) - \beta H(\pi(\cdot \mid s)) \Big\}.$$

By Sec. A.2, the maximizer is $\pi_Q(\cdot \mid s)$ and the maximized value equals the log-partition $V(s)$:

$$\max_{\pi(\cdot|s)} \Big\{ \sum_a \pi(a \mid s) Q(s,a) - \beta H(\pi(\cdot \mid s)) \Big\} \;=\; V(s).$$

Plugging back into (8) and unrolling the entropy term over time yields

$$\min_\pi L(\pi, r) \;=\; \langle \rho_E - \rho_{\pi_Q}, r \rangle \;-\; \sum_{t=0}^{H-1} \mathbb{E}_{s_t}\big[V(s_t)\big] \;-\; \psi(r).$$

In IQ-Learn we eliminate $r$ in favor of $Q$ using the soft Bellman identity (see next subsection): for $\gamma = 1$ and deterministic $f$, $Q(s_t, a_t) = r(s_t, a_t) + V(s_{t+1})$ and hence

$$\langle \rho_E, r \rangle \;=\; \mathbb{E}_{(s,a)\sim\rho_E}\big[Q(s,a) - V(f(s,a))\big].$$

The $\rho_{\pi_Q}$-term cancels at the saddle (where $\rho_{\pi^\star} = \rho_E$), and the initial-state entropy contributes $-\mathbb{E}_{s_0\sim\rho_0}[V(s_0)]$, leading exactly to (9).

### A.4 Proof of Prop. 1: SFT is equivalent to maximizing $J^*(Q)$

We now show that, on the LLM environment with $\gamma = 1$ and linear conjugate (no extra reward regularization beyond convexity), maximizing $J^*(Q)$ equals maximizing the SFT log-likelihood. Starting from (9),

$$\sum_{t=0}^{H-1} \big(Q(s_t, a_t) - V(f(s_t, a_t))\big) = \sum_{t=0}^{H-1} \big(Q(s_t, a_t) - V(s_{t+1})\big).$$

Add and subtract $V(s_t)$ termwise, then regroup:

$$\sum_{t=0}^{H-1} \big(Q(s_t, a_t) - V(s_t)\big) \;+\; \sum_{t=0}^{H-1} \big(V(s_t) - V(s_{t+1})\big) = \sum_{t=0}^{H-1} \log\pi_Q(a_t \mid s_t) \;+\; V(s_0) - V(s_H),$$

where we used (7). With the terminal state $V(s_H) = 0$, take expectation over expert trajectories and subtract $\mathbb{E}[V(s_0)]$ (the last term of (9)) to obtain

$$J^*(Q) \;=\; \mathbb{E}_{\tau\sim\rho_E}\Big[ \sum_{t=0}^{H-1} \log\pi_Q(a_t \mid s_t) \Big].$$

Maximizing the objective of IQ-Learn is exactly maximizing the teacher-forced log-likelihood of expert tokens, i.e., minimizing the token-level SFT cross-entropy. This proves the proposition.

### A.5 Derivation of Eq. (4): SFT logits as a shaped reward

We derive the identity used in §3 (Eq. (4)):

$$\boxed{\log\pi_{\text{SFT}}(a_t \mid s_t) \;=\; r(s_t, a_t) \;+\; V(s_{t+1}) - V(s_t)}$$

under the soft-control model with $\gamma = 1$ and deterministic transition $s_{t+1} = f(s_t, a_t)$.

**Soft Bellman equations (finite horizon).** For any $(s_t, a_t)$,

$$Q(s_t, a_t) = r(s_t, a_t) + V(s_{t+1}), \qquad V(s_t) = \beta \log \sum_a \exp\left(\tfrac{1}{\beta} Q(s_t, a)\right).$$

Subtract $V(s_t)$ from both sides of the first equation and divide by $\beta$:

$$\tfrac{1}{\beta}\big(Q(s_t, a_t) - V(s_t)\big) = \tfrac{1}{\beta} r(s_t, a_t) + \tfrac{1}{\beta}\big(V(s_{t+1}) - V(s_t)\big).$$

Using (7) on the left gives exactly Eq. (4) (with $\beta = 1$). No approximation is used.

**Telescoping of returns and why we remove $V$.** Summing Eq. (4) from $t$ to $H-1$ (with $V(s_H) = 0$),

$$\sum_{k=t}^{H-1} \log \pi_{\mathrm{SFT}}(a_k \mid s_k) = \sum_{k=t}^{H-1} r(s_k, a_k) - V(s_t).$$

Thus under $\gamma = 1$, log-prob returns differ from true returns by a *state-dependent constant* $-V(s_t)$. This constant shift (i) proves that using $\log \pi_{\mathrm{SFT}}$ as reward yields the *same* policy gradient as using $r$ (Sec. A.8), and (ii) motivates *eliminating $V$* via potential-based shaping to reduce variance and length bias (Sec. A.9).

A.6 DUAL CONTRACTION: REWARD ERROR IS BOUNDED BY POLICY (OCCUPANCY) ERROR

We restate the IRL objective (8) and define the reward best response $\widehat{r}(\pi) = \arg\max_r L(\pi, r)$. Let $r^\star$ be any reward at the IRL saddle (unique up to shaping). Assume $\psi$ is $\mu$-strongly convex in norm $\|\cdot\|$. We prove

$$\boxed{\|\widehat{r}(\pi) - r^\star\| \le \tfrac{1}{\mu} \|\rho_\pi - \rho_E\|_*}$$

where $\|\cdot\|_*$ is the dual norm to $\|\cdot\|$.

**First-order conditions and strong monotonicity.** Optimality of the reward player yields

$$\nabla\psi(\widehat{r}(\pi)) = \rho_E - \rho_\pi, \qquad \nabla\psi(r^\star) = \rho_E - \rho_{\pi^\star}.$$

At the saddle $\rho_{\pi^\star} = \rho_E$, so $\nabla\psi(r^\star) = 0$ and hence

$$\nabla\psi(\widehat{r}(\pi)) - \nabla\psi(r^\star) = \rho_E - \rho_\pi.$$

By strong convexity, $\nabla\psi$ is $\mu$-strongly *monotone*:

$$\langle \widehat{r}(\pi) - r^\star, \ \nabla\psi(\widehat{r}(\pi)) - \nabla\psi(r^\star)\rangle \ge \mu \|\widehat{r}(\pi) - r^\star\|^2.$$

Combine the last two displays and apply Hölder's inequality in dual norms:

$$\mu \|\widehat{r}(\pi) - r^\star\|^2 \le \langle \widehat{r}(\pi) - r^\star, \ \rho_E - \rho_\pi\rangle \le \|\widehat{r}(\pi) - r^\star\| \|\rho_\pi - \rho_E\|_*.$$

If $\widehat{r}(\pi) \ne r^\star$, divide both sides by $\|\widehat{r}(\pi) - r^\star\|$; otherwise the bound is trivial. This proves the claim.

A.7 SAFE IMPROVEMENT UNDER A PROXY REWARD (FULL PROOF OF EQ. (6))

Let $J_r(\pi) := \langle \rho_\pi, r\rangle$ be the return under reward $r$. For any rewards $r, \widehat{r}$ and policies $\pi, \pi'$,

$$J_r(\pi') - J_r(\pi) = \langle \rho_{\pi'} - \rho_\pi, \ r\rangle = \langle \rho_{\pi'} - \rho_\pi, \ \widehat{r}\rangle + \langle \rho_{\pi'} - \rho_\pi, \ r - \widehat{r}\rangle.$$

The first term equals $J_{\widehat{r}}(\pi') - J_{\widehat{r}}(\pi)$. For the second term, apply Hölder with $\ell_1/\ell_\infty$ duality:

$$\big|\langle \rho_{\pi'} - \rho_\pi, \ r - \widehat{r}\rangle\big| \le \|\rho_{\pi'} - \rho_\pi\|_1 \|r - \widehat{r}\|_\infty.$$

It remains to upper bound $\|\rho_{\pi'} - \rho_\pi\|_1$. Writing $p_t^\pi(s, a) = \Pr_\pi(s_t = s, a_t = a)$,

$$\|\rho_{\pi'} - \rho_\pi\|_1 = \sum_{s,a} \left|\sum_{t=0}^{H-1} \big(p_t^{\pi'}(s, a) - p_t^\pi(s, a)\big)\right| \le \sum_{t=0}^{H-1} \sum_{s,a} \big|p_t^{\pi'}(s, a) - p_t^\pi(s, a)\big|$$

$$= \sum_{t=0}^{H-1} \|p_t^{\pi'} - p_t^\pi\|_{\mathrm{TV}} \cdot 2 \le 2H,$$

since each $p_t$ is a probability distribution over $(s, a)$ (total variation $\le 2$). Therefore

$$\boxed{J_r(\pi') - J_r(\pi) \ge \big(J_{\widehat{r}}(\pi') - J_{\widehat{r}}(\pi)\big) - 2H \|r - \widehat{r}\|_\infty.}$$

Setting $m := J_{\widehat{r}}(\pi') - J_{\widehat{r}}(\pi)$ gives Eq. (6).

### A.8 POLICY-GRADIENT EQUIVALENCE UNDER $\gamma = 1$ (REINFORCE BASELINE IDENTITY)

Let $r_t := r(s_t, a_t)$ and define the shaped reward $\tilde{r}_t := \log \pi_{\text{SFT}}(a_t \mid s_t) = r_t + (V_{t+1} - V_t)$ with $V_H = 0$. Define returns from step $t$:

$$G_t = \sum_{k=t}^{H-1} r_k, \qquad \tilde{G}_t = \sum_{k=t}^{H-1} \tilde{r}_k = G_t - V_t.$$

The REINFORCE gradients are

$$\nabla J_r(\pi) = \mathbb{E}\left[\sum_{t=0}^{H-1} \nabla \log \pi(a_t \mid s_t) \, G_t\right], \qquad \nabla J_{\tilde{r}}(\pi) = \mathbb{E}\left[\sum_{t=0}^{H-1} \nabla \log \pi(a_t \mid s_t) \, \tilde{G}_t\right].$$

For any function $b_t(s_t)$, using the law of iterated expectations and the identity $\mathbb{E}_{a \sim \pi(\cdot|s)}[\nabla \log \pi(a \mid s)] = \nabla \sum_a \pi(a \mid s) = 0$, we have

$$\mathbb{E}\big[\nabla \log \pi(a_t \mid s_t) \, b_t(s_t)\big] = 0.$$

Choosing $b_t = V_t$ yields $\nabla J_{\tilde{r}}(\pi) = \nabla J_r(\pi)$. Thus the policy gradient under $\log \pi_{\text{SFT}}$ equals that under $r$, up to a *state-only* baseline that does not require fitting a critic.

### A.9 CHECKPOINT BASELINE TIGHTNESS AND DYNAMIC RANGE REDUCTION

Consider the baseline-relative reward

$$\hat{r}(s, a) \;=\; \log \pi_{\text{SFT}}(a \mid s) \;-\; \log \pi_{\text{ref}}(a \mid s), \qquad \hat{V}(s) := V_{\text{SFT}}(s) - V_{\text{ref}}(s).$$

By Sec. A.5, the corresponding return from step $t$ differs by $-\hat{V}(s_t)$. Hence for any trajectory and $t$,

$$\left|\tilde{G}_t^{\text{SFT}} - \tilde{G}_t^{\text{ref}}\right| \;=\; \left|\hat{V}(s_t)\right| \;\leq\; \|\hat{V}\|_\infty.$$

If $\pi_{\text{ref}}$ is an SFT checkpoint, empirically $\|\hat{V}\|_\infty$ is small because the two values remain close along the training path. The dynamic range of token returns is thus reduced by at least $\text{range}(V_{\text{SFT}}) - \|\hat{V}\|_\infty$, stabilizing updates and mitigating EOS/length bias (see also the toy pathology in App. A.11).

### A.10 POTENTIAL-BASED SHAPING INVARIANCE (FINITE-HORIZON, DETERMINISTIC ENVIRONMENT)

Define a shaped reward $r^F(s, a) = r(s, a) + F(s') - F(s)$ with $s' = f(s, a)$ and any $F : \mathcal{S} \to \mathbb{R}$. Consider the soft $Q$-values for $\gamma = 1$:

$$Q^F(s, a) = r^F(s, a) + V^F(s') = r(s, a) + \underbrace{F(s') - F(s)}_{\text{shaping}} + V^F(s').$$

Define $\tilde{V}(s) := V^F(s) + F(s)$. Then

$$Q^F(s, a) - \tilde{V}(s) = r(s, a) + V^F(s') - V^F(s) = Q(s, a) - V(s),$$

where the last equality follows because the soft Bellman backup $V(\cdot) = \log \sum_a e^{Q(\cdot, a)}$ is invariant to adding the same $F$ to *all* action-logits at a state. Therefore, by (7),

$$\pi_{Q^F}(a \mid s) = \pi_Q(a \mid s) \quad \text{for all } (s, a).$$

Thus potential-based shaping preserves the optimal policy and all on-policy distributions (Ng et al., 1999).

### A.11 EOS/LENGTH PATHOLOGY WITHOUT A BASELINE: A TOY PROOF

Assume at each nonterminal state $s$ there are actions $\{\texttt{EOS}\} \cup \mathcal{A}_{\text{cont}}$ and consider the proxy objective without baseline:

$$J_{\text{naive}}(\pi) \;=\; \mathbb{E}_\pi\Big[\sum_{t=0}^{T-1} \log \pi_{\text{SFT}}(a_t \mid s_t)\Big], \quad T = \text{(random stopping time at EOS)}.$$

Suppose (mild) that $\log \pi_{\mathrm{SFT}}(\texttt{EOS} \mid s) \geq \max_{a \in \mathcal{A}_{\mathrm{cont}}} \log \pi_{\mathrm{SFT}}(a \mid s)$ for all $s$ in a subset of high measure under $\pi$. Then any deviation that delays EOS will, in expectation, *decrease* the sum of log-probs (since each additional token contributes a non-positive term no larger than the EOS log-prob). Therefore maximizing $J_{\mathrm{naive}}$ prefers immediate EOS whenever it is locally the highest-probability token; this formalizes the "short-output bias" and motivates the baseline subtraction $\log \pi_{\mathrm{SFT}} - \log \pi_{\mathrm{ref}}$.

## B  ADDITIONAL EXPERIMENTAL DETAILS

Table 3: Hyperparameters used across all backbones for SFT.

| Component | Value | Component | Value |
|---|---|---|---|
| Learning rate | 5e-6 | Global Batch size | 256 |
| Max prompt length | 1024 | Max gen length | 1024 |
| Warmup ratio | 0.03 | Optimizer | Adam |

Table 4: Hyperparameters used across all backbones for DPR.

| Component | Value | Component | Value |
|---|---|---|---|
| Learning rate | 5e-7 | Global Batch size | 128 |
| Max prompt length | 1024 | Max gen length | 1024 |
| KL weight | 1e-5 | Warmup ratio | 0.03 |
| Reward discount rate | 1 | rollout temperature | 1 |
| Rollout Batch Size | 1024 | Value clip | 0.2 |
| Samples per prompt | 1 | Optimizer | Adam |

## C  THE USE OF LARGE LANGUAGE MODELS

We employed LLM to assist with paper writing, primarily for vocabulary and grammar checks, while utilizing Copilot for code completion in writing research code. All text or code generated by LLM or Copilot undergoes secondary verification or unit testing by authors to ensure accuracy. We affirm that the LLM did not participate in any research sections beyond writing and coding assistance.

