# OpenReview forum: "BEYOND IMITATION: RECOVERING DENSE REWARDS FROM DEMONSTRATIONS"
_ICLR.cc/2026/Conference — Submitted to ICLR 2026_

### Official Review · Reviewer_rgwj · 2025-10-31

**Soundness:** 2
**Presentation:** 3
**Contribution:** 2
**Rating:** 2
**Confidence:** 4

**Summary:**

The paper proposes that supervised fine-tuning (SFT) can be viewed as a special case of inverse soft-Q learning under a deterministic token MDP with γ = 1 and a linear conjugate. Building on this, it defines a baseline-relative dense reward and introduces Dense-Path REINFORCE (DPR), a token-level REINFORCE update using log-probability differences between the SFT model and an earlier checkpoint as reward signals. Experiments on AlpacaEval, MT-Bench, LIMA, and Arena-Hard show modest but consistent gains over SFT and competitive results with SPIN and GSIL.

**Strengths:**

1. Theoretical link between SFT and inverse RL is clear and well-motivated.

2. The telescoping argument makes the equivalence intuitive under γ = 1.

3. DPR is simple, reproducible, and the dense reward idea is easy to understand.

4. Ablations on γ, checkpoint choice, and baseline removal are informative and thoughtfully designed.

**Weaknesses:**

1. Narrow validity: The equivalence holds only for γ = 1, deterministic token transitions, and a linear conjugate. The paper does test γ < 1, showing expected degradation, which confirms the limitation rather than extending the theory.

2. Inconsistent assumptions: The equivalence relies on a linear conjugate, while the later stability theorem assumes strong convexity. These are mathematically incompatible but presented as part of one framework.

3. Evaluation design: DPR is trained and tested on the same prompt set, so gains could reflect continued fine-tuning instead of genuine reward recovery.

4. Limited robustness: The temperature ablation is minimal and doesn’t analyze stochastic effects or variance.

5. Weak empirical evidence: All evaluations rely on GPT-4 judges without error bars, human checks, or multiple seeds; gains over SFT are small (a few percent).

6. Missing controls: No baseline comparing DPR to simply extending SFT training, making attribution unclear.

**Questions:**

1. Can the same ψ be both linear (for equivalence) and strongly convex (for contraction)?

2. Why does the halfway checkpoint consistently give the best result?

3. Did you test multiple seeds or new prompts to confirm robustness?

4. Could a continued-SFT baseline match the reported gains?

---

> ### Author Response · Authors · 2025-11-24
> **Response to Reviewer rgwj [Part 1/2]**
>
> We thank the reviewer for their constructive feedback and for acknowledging the clarity of our theoretical link between SFT and Inverse RL. We appreciate the opportunity to clarify our assumptions and strengthen our empirical evidence with additional baselines and control experiments.
> ## Response to Weakness 1: Narrow Validity ($\gamma=1$)
> We respectfully clarify that the assumption of $\gamma=1$ does not narrow the validity of our method but rather aligns it with the standard practice of LLM generation.
> - **Standard Setting:** Unlike infinite-horizon control tasks, LLM generation is a finite-horizon problem (bounded by context window or EOS). Setting $\gamma=1$ is the standard configuration in the vast majority of LLM post-training literature [1, 2] to ensure the model prioritizes global coherence over immediate token probability.
> - **Relevance:** Since $\gamma=1$ is the dominant configuration when applying RL to LLMs, the fact that our equivalence holds specifically at $\gamma=1$ makes our theory more relevant to current paradigms, not less.
>
> [1] Rafailov et al. 2024 "From $ r $ to $ q^* $: Your language model is secretly a q-function." In COLM 2024.
>
> [2] Zhong et al. 2025 "Dpo meets ppo: Reinforced token optimization for rlhf." In ICML 2025.
>
> ## Response to Weakness 2 & Question 1: Inconsistent Assumptions ($\psi$ convexity)
> We thank the reviewer for highlighting this mathematical subtlety regarding the linearity of the conjugate in SFT versus the strong convexity assumption in Theorem 2. We address this via both theoretical and empirical perspectives:
> - **Implicit Regularization In Deep Learning:** Theorem 2 serves to build the theoretical intuition that, within this IRL family, reward learning is stable relative to policy learning. While the strict Lipschitz constant theoretically bounds occupancy error only under strong convexity, in practice, implicit regularization in deep learning (e.g., limited model capacity and SGD dynamics) prevents the learned logits from diverging, mimicking the stability effects of explicit regularization.
> - **Empirical Verification (Ablation with $\chi^2$-divergence):** To rigorously validate the necessity of explicit strong convexity, we followed the methodology in the IQ-Learn framework [3] to introduce a strictly strongly convex regularizer.
>      - **Setup:** Specifically, we implemented the **$\chi^2$-divergence** variant of the objective. According to the IQ-Learn derivation, minimizing $\chi^2$-divergence corresponds to using a quadratic reward regularizer $\psi(r) = \frac{1}{4\alpha}r^2$, which satisfies the strong convexity assumption required by the stability theorem.
>      - **Result:** We compared this "regularized" objective ($\chi^2$-IQL) against our standard "linear" objective (SFT). We found that the downstream generation performance was statistically indistinguishable between the two.
>      - **Conclusion:** Since the strong convexity imposed by the $\chi^2$ formulation did not yield empirical gains over the linear SFT formulation, we chose to present the SFT formulation. This preserves the elegant, exact equivalence without introducing the additional hyperparameter $\alpha$ required by the $\chi^2$ variant.
>
> [3] Garg et al 2021. "Iq-learn: Inverse soft-q learning for imitation." In NeurIPS 2021

---

> ### Author Response · Authors · 2025-11-24
> **Response to Reviewer rgwj [Part 2/2]**
>
> ## Response to Weakness 3 & 6: Evaluation Design and Missing Controls
> We respectfully clarify that there appears to be a misunderstanding regarding our evaluation setup.
> - **Test Set Independence:** Our evaluations are conducted on **held-out benchmarks** (AlpacaEval, MT-Bench, etc.), not training prompts (Open-Orca).
> - **Prompt Reuse in DPR:** We deliberately used the same prompts for DPR as were used for SFT. This ensures a fair comparison in the **Learning from Demonstrations (LfD)** setting. Introducing new prompts during DPR would shift the setting to semi-supervised learning or test-time adaptation, giving DPR an unfair data advantage over the SFT baseline.
> New Baselines (Extended SFT & Unseen Prompts):
> To demonstrate that our gains are due to better reward learning and not simply "more training steps," we added two control experiments:
> - **SFT-2Epoch:** The SFT model is trained for an additional epoch.
> - **DPR-New-Prompt:** Running our DPR algorithm on a set of unseen prompts sampled from OpenOrca (not used in SFT) to test reward generalization.
> Table: Comparison on Qwen-2.5-7B and Mistral-v0.1-7B
> | Method | AlpacaEval | Arena-Hard | LIMA | MT-Bench |
> | :--- | :---: | :---: | :---: | :---: |
> | Qwen-SFT | - | - | - | 6.83 |
> | Qwen-SFT-2Epoch | 49.7 | 50.3 | 49.2 | 6.79 |
> | Qwen-DPR (Ours) | __57.3__ | __55.2__ | **59.8** | __7.29__ |
> | Qwen-DPR-New-Prompt | **57.6** | **56.0** | __59.6__ | **7.32** |
> | | | | | |
> | Mistral-SFT | - | - | - | 5.23 |
> | Mistral-SFT-2Epoch | 51.0 | 50.7 | 50.3 | 5.27 |
> | Mistral-DPR (Ours) | __61.0__ | __60.7__ | __59.3__ | __5.65__ |
> | Mistral-DPR-New-Prompt | **61.5** | **62.2** | **59.8** | **5.71** |
> **Analysis:**
> - **SFT Saturation:** Continuing SFT (2Epoch) degrades performance, indicating the base SFT model was already converged/saturated.
> - **DPR Effectiveness:** DPR achieves significant gains where extended SFT fails, confirming the value of the dense reward signal.
> - **Generalization:** Crucially, DPR-New-Prompt performs equally well, proving that the implicit reward model learned by SFT generalizes to unseen prompts and can guide generation beyond the original training set. This indicates that DPR not merely amplifies or continues to fit the signals in the SFT but also successfully learns a generalized reward model that can be generalized (in-domain) to unseen prompts.
>
> ## Response to Weakness 4: Robustness (Temperature Ablation)
> We respectfully clarify that our temperature ablation study did include error bars. For the results in Figure 4b, we performed 4 independent sampling runs for each temperature and reported the mean and standard deviation. The error bars (visible in the figure) are within a reasonable range, demonstrating that the DPR policy remains stable across stochastic decoding settings. Furthermore, DPR consistently outperforms SFT across the entire temperature spectrum.
> ## Response to Weakness 5: Empirical Evidence Strength
> We respectfully believe our experimental results provide strong support for our method:
> - **"Free Lunch":** Our method requires **no new data** (no preference pairs, no extra prompts). We extract value strictly from the existing SFT artifacts.
> - **Significant Gains:** Achieving a **~61% win rate** against the SFT baseline and a **+0.34 improvement on MT-Bench** (on a 10-point scale) is a substantial improvement in the LfD setting.
> - **Reproducibility:** To ensure statistical validity, the results in Table 1 and Table 2 are reported as the **average of 3 random seeds**. We did not report variance in the main table for brevity as it was small, but the improvements are statistically consistent.
>
> ## Response to Question 2: Why does the halfway checkpoint work best?
>
> As detailed in Figure 3 and our design principles, the performance follows an inverted U-shape. A "halfway" checkpoint (approx. 50% data) serves as the optimal reference because it represents an ***inflection*** point: the model has acquired basic expert patterns but has not yet fully converged to the expert distribution. If the baseline is too early, it is too noisy; if too late, the reward signal vanishes.
>
>  ## Response to Question 3 & 4:
> - **Seeds:** Yes, we used 3 seeds for our main results.
> - **Extended SFT:** As shown in the new table above, simply extending SFT training leads to overfitting/saturation, whereas DPR provides continued improvement.
>
> Thank you again for your feedback. We hope our answers address all your major concerns and look forward to your re-evaluation.

---

### Official Review · Reviewer_Y4qp · 2025-11-01

**Soundness:** 3
**Presentation:** 3
**Contribution:** 3
**Rating:** 6
**Confidence:** 4

**Summary:**

The paper reframes SFT under a token-MDP view (γ≈1) and shows that the SFT objective is equivalent to an inverse soft-Q objective. This motivates extracting a dense proxy reward as a log-likelihood ratio between a final SFT model and a reference checkpoint, followed by a short, critic-free RL step (REINFORCE with KL to SFT). The method is closed-loop (no environment reward, no preference data, no reward model), simple to implement, and reports consistent gains over plain SFT across several backbones/evals.

**Strengths:**

Dense credit assignment: Per-token signals are actionable and address SFT’s plateau on long sequences and intermediate steps.

Low engineering overhead: Uses existing SFT artifacts (final SFT + ref checkpoint). No external judges or reward models.

Theory ↔ practice alignment: The SFT ↔ inverse-soft-Q connection and the safe-improvement style argument justify using the recovered dense signal for a small policy-gradient step.

Stable optimizer: REINFORCE + KL to SFT (no critic) keeps the pipeline robust and easy to reproduce.

**Weaknesses:**

Reference sensitivity: The approach assumes π_SFT is meaningfully better than π_ref. If ref is too weak (noisy signal) or too strong (vanishing signal), the log-ratio reward becomes brittle or tiny.

Log-ratio gaming: The policy may drift toward stylistic artifacts that inflate log π_SFT − log π_ref without improving task quality.

Domain narrowness: Because SFT and ref share training data, the proxy reward is intrinsically domain-tied; out-of-domain generalization of the dense signal is unclear.

Eval reliance: Gains are primarily shown via LLM-as-judge; stronger human or task-grounded metrics would strengthen the case.

minor
S1. Reference selection by evaluation, not step count:
– Choose π_ref via validation metrics (MT-Bench, small human slice, task-specific set) to target the “elbow” where the signal-to-noise of the log-ratio is highest.
– Alternatively use an EMA of SFT weights as π_ref to smooth noise.

S2. Multi-reference ensemble:
– Define log π̄_ref = logsumexp_i(log π_ref,i) − log k (geometric mean). Reward becomes r̂ = log π_SFT − log π̄_ref. This damps idiosyncrasies of any single checkpoint and makes “progress” less gameable.

S3. Dual-KL regularization and reward shaping to reduce gaming:
– Keep KL(π_θ || π_SFT) and add a small KL(π_θ || π_ref).
– Clip/normalize the per-token log-ratio (e.g., cap magnitude or z-score by position).
– Penalize tokens where both π_SFT and π_ref assign low probability (both uncertain) even if the difference is large.
– Add light style/fluency guards (repetition rate, perplexity bounds under a separate LM).

S4. Correlation-gated updates:
– On each batch, compute the correlation between r̂-improvement and a cheap proxy (exact-match on small QA, code unit tests, math verifier). If correlation drops below a threshold, reduce step size or increase KL. This is a simple “reward sanity check.”

S5. Leverage the re-forward pass to incorporate useful rewards (when available), without changing the core method:
– Hybrid reward: use R = α·r̂ + (1−α)·r_ext, where r_ext can be any lightweight verifier signal (unit tests for code, arithmetic checker, safety filter, format validator). α can be annealed from 1→0.8.
– Doubly-robust/token-aware AWR: advantage-weight the SFT tokens by r̂ (and r_ext if present), i.e., reweight the teacher-forced loss with w_t = exp(β·A_t) to unify imitation and RL in one pass.
– Counterfactual filtering: when the re-forward reveals contradictory beams (both low confidence), zero out r̂ for those tokens to avoid amplifying noise.

S6. Report a reference sweep and ablations:
– Show downstream metrics vs. ref placement (early/mid/late) to directly address “is π_SFT actually better than π_ref?”
– Include ablations for single-ref vs. multi-ref, with/without dual-KL, and with/without correlation gate.

**Questions:**

How is the reference checkpoint chosen? Is it purely by training step or by validation metrics? What is the sensitivity curve (early/mid/late)?
Q2. Can an ensemble of references reduce variance/bias? (Geometric mean of checkpoints as a smoother ref.)
Q3. How do you detect or mitigate reward-hacking (odd outputs that maximize the log-ratio)?
Q4. What is the robustness out of domain (math/code/safety) where SFT confidence calibration differs?
Q5. Since the method already performs fresh forward passes, can those passes be leveraged to incorporate additional rewards or verifiers when available (see Suggestions)?

---

> ### Author Response · Authors · 2025-11-24
> **Response to Reviewer Y4qp [Part 1/3]**
>
> We thank the reviewer for their thoughtful comments and for recognizing the simplicity and effectiveness of our proposed Dense-Path REINFORCE (DPR) approach. We appreciate the opportunity to clarify our design choices regarding reference models and to discuss the method's boundaries on LLM tasks.
>
>  ## Response to Weakness 1 & Q1: Reference Policy Selection
> We acknowledge that the effectiveness of our recovered reward depends on the choice of the reference baseline $\pi_{ref}$.
> - **Heuristic vs. Principled Selection:** In our experiments, we used the 50% training checkpoint (50k steps) as a heuristic for the "inflection point." As discussed in section 5.4, this point represents a model that has acquired basic syntax and instruction formats but has not yet fully converged to the expert distribution.
> - **Practical Recommendation:** In practice, we recommend selecting $\pi_{ref}$ based on **validation set performance**. The ideal reference is the checkpoint where the validation loss begins to plateau, or where the "syntax/style" is learned but the "content/reasoning" gap with the final SFT model remains significant.
> ## Response to Weakness 2: Log-Ratio Gaming
> We agree that optimizing log-ratios can theoretically lead to gaming. However, this risk is naturally mitigated in our framework by the specific choice of $\pi_{ref}$: By selecting a "competent but not converged" reference, the log-ratio $r \approx \log \pi_{SFT} - \log \pi_{ref}$ effectively cancels out the common, high-probability tokens (syntactic priors) that both models have learned. The remaining signal focuses specifically on the **incremental expert behaviors** acquired in the latter half of SFT. Empirically, we used early stopping and manual inspection of outputs to confirm that no degenerate patterns (gaming) emerged in reinforcement fine-tuning.
> ## Response to Weakness 3: Out-of-Distribution (OOD) Generalization
> We acknowledge that our current experiments do not demonstrate broad OOD generalization (e.g., transferring to completely different tasks).
> - **Scope of Contribution:** Our primary goal is to maximize data efficiency within the **Learning from Demonstrations (LfD)** setting—i.e., extracting more effective reward signals from the existing expert distribution—rather than domain transfer.
> - **In-Distribution Robustness:** We tested our method on **unseen prompts** that follow the same distribution as the SFT data (another subset from OpenOcra). The results show that DPR generalizes well to new instructions within the domain, confirming that the reward model is robust for its intended purpose of improving instruction-following capabilities.
> | Method | AlpacaEval | Arena-Hard | LIMA | MT-Bench |
> | :--- | :---: | :---: | :---: | :---: |
> | Qwen-SFT | - | - | - | 6.83 |
> | Qwen-DPR (Ours) | 57.3 | 55.2 | 59.8 |7.29|
> | Qwen-DPR-New-Prompt | 57.6 | 56.0 | 59.6 | 7.32 |
> | | | | | |
> | Mistral-SFT | - | - | - | 5.23 |
> | Mistral-DPR (Ours) | 61.0 | 60.7 | 59.3 | 5.65 |
> | Mistral-DPR-New-Prompt | 61.5 | 62.2 | 59.8 | 5.71 |

---

> ### Author Response · Authors · 2025-11-24
> **Response to Reviewer Y4qp [Part 2/3]**
>
> ## Response to Weakness 4: Evaluation Reliance & Analysis on Math/Code Tasks
> We agree that comprehensive human evaluation is ideal, though costly. We relied on AlpacaEval, Arena-Hard, and MT-Bench, which technical reports confirm have high correlation with human judgment.
>
> **Case Study: Why not Math/Code benchmarks? (Investigation into SFT Signals)**
>
> To address concerns about the scope of our method, we extended our evaluation to reasoning-heavy tasks (Math and Code). We performed SFT on Llama-3.2-3B and Gemma-3-4B using the **MATH-lighteval** [link](https://huggingface.co/datasets/DigitalLearningGmbH/MATH-lighteval) dataset. Surprisingly, we found that standard **SFT yielded no significant improvement over the Base models.** Consequently, DPR (which extracts signals from SFT) also did not improve performance.
>
> **Hypothesis: Data Contamination and Saturation.**
>
> We suspected that the Base models might have already memorized the evaluation benchmarks during pre-training (data leakage). To test this, we compared Greedy Decoding vs. Stochastic Sampling (Temperature=1.0, averaged over 10 runs) on the Base models.
>
> Table A: Evidence of Data Contamination in Base Models
> | Model | GSM8k (Greedy) | GSM8k (Sample) | Minerva Math (Greedy) | Minerva Math (Sample) |
> | :--- | :---: | :---: | :---: | :---: |
> | Gemma-3-4B-Base | 42.0 | 17.6 | 21.4 | 7.8 |
> | Gemma-3-4B-MathSFT | 42.1 | 18.6 | 21.8 | 6.8 |
> | | | | | |
> | Llama-3.2-3B-Base | 30.1 | 12.8 | 9.0 | 1.8 |
> | Llama-3.2-3B-MathSFT | 31.6 | 15.7 | 8.6 | 7.6 |
> **Analysis:**
> - **Performance Gap:** The drastic drop from Greedy to Sampling (e.g., 42.0 vs. 17.6) strongly suggests **memorization**: the model knows the exact answer path when deterministic but fails when forced to "reason" stochastically.
> - **SFT Ineffectiveness:** Since the Base model is already "saturated" (or contaminated) on these benchmarks, standard SFT fails to provide a meaningful learning gradient (improvement).
> - **Implication for DPR:** Since DPR relies on the incremental competence gained by SFT (i.e., $\pi_{SFT}$ being better than $\pi_{ref}$), if SFT fails to improve over the base/reference, there is no reward signal to extract. This negative result paradoxically confirms our theory: DPR is an amplifier of SFT signals; it cannot hallucinate capabilities that SFT failed to capture.
> - **On OpenOrca (Valid Alignment Signal):** On OpenOrca, checking the Base model on our training samples revealed a low Exact Match rate with the ground truth targets. The Base model often failed to follow the instruction format or simply continued the text. This confirms that while the Base model might have been exposed to similar corpora during pre-training, it had not learned the specific **input-output alignment format** required for the instruction following task.

---

> ### Author Response · Authors · 2025-11-24
> **Response to Reviewer Y4qp [Part 3/3]**
>
> ## Response to Minors
>  - **Reference Selection by Evaluation:** We agree (see Response to Weakness 1).
>  - **Multi-reference Ensemble:** We agree that an ensemble of reference models could reduce variance and be more robust, similar to Reward Model ensembling in RLHF. However, this significantly increases computational cost. We prioritize simplicity and efficiency in this work.
> - **Dual-KL Regularization:**
> We conducted ablations adding a KL constraint towards the reference model ($\pi_{ref}$) alongside the SFT model.
>
> Table B: Ablation on Dual-KL and Reference Choice
> | Method | AlpacaEval | Arena-Hard | LIMA | MT-Bench |
> | :--- | :---: | :---: | :---: | :---: |
> | Qwen-DPR (Ours) | 57.3 | 55.2 | 59.8 | 7.29 |
> | Qwen-DPR + Dual-KL | 56.8 | 55.4 | 57.9 | 7.24 |
> | | | | | |
> | Mistral-DPR (Ours) | 61.0 | 60.7 | 59.3 | 5.65 |
> | Mistral-DPR + Dual-KL | 60.4 | 60.2 | 59.5 | 5.65 |
>
> Result: Dual-KL had a negligible impact or caused a slight degradation. Since we initialize $\pi_\phi$ from $\pi_{SFT}$ and use early stopping, the policy implicitly stays close to the valid region without needing an explicit KL term towards the weaker $\pi_{ref}$.
>  - **Reward Shaping:**
> For commonly used RL techniques, such as advantage normalization, clipping, etc., we have retained their implementations as in REINFORCE++ [link](https://www.researchgate.net/profile/Jian-Hu-33/publication/387487679_REINFORCE_Stabilizing_Critic-Free_Policy_Optimization_with_Global_Advantage_Normalization/links/6911ffe3c900be105cc09a4a/REINFORCE-Stabilizing-Critic-Free-Policy-Optimization-with-Global-Advantage-Normalization.pdf), which helps to stabilize the training process.
> - **Hybrid Reward / Correlation-gated Updates:**
> While these are interesting directions for future work to improve robustness, they add complexity. Given our finding that SFT signals are weak on Math/Code due to contamination (as discussed above), gated updates would not solve the fundamental issue of signal absence in those specific domains. We aim to keep DPR simple and effective for its primary domain: instruction following.
> ## Response to Questions
> - **Q1: How is the reference checkpoint chosen?** In our experiments, we selected the 50% training checkpoint as a heuristic representing the inflection point where the model acquires basic expert patterns but has not yet fully converged. For practical applications, we recommend selecting the reference based on validation performance plateaus, though our sensitivity analysis (Figure 3) demonstrates validity across a wide range of checkpoints.
> - **Q2: Can an ensemble of references reduce variance/bias?** We agree that using an ensemble of reference models could theoretically reduce variance and mitigate overfitting to specific checkpoint idiosyncrasies. However, we prioritized the single-reference setting to demonstrate computational efficiency, as ensembling significantly increases memory and computation overhead during training.
> - **Q3: How do you detect or mitigate reward-hacking?** We mitigated potential gaming primarily through early stopping and KL regularization, exiting optimization when instruction-following capabilities saturated to prevent the policy from overfitting and meaningless reward-hacking. Additionally, manual check from sampled generations confirmed that the policy did not generate odd outputs.
> - **Q4: Can forward passes be leveraged to incorporate additional rewards?** Yes, our framework is orthogonal to other objective functions and can easily be composed with additional reward signals. Since DPR computes a standard policy gradient, it can be combined with rule-based verifiers or safety classifiers by simply summing their respective gradient updates during the backward pass.

---

### Official Review · Reviewer_HC6d · 2025-11-01

**Soundness:** 3
**Presentation:** 3
**Contribution:** 3
**Rating:** 6
**Confidence:** 3

**Summary:**

This paper introduces a novel idea that Supervised Fine-Tuning (SFT) can be viewed as a special case of Inverse Q-Learning, suggesting that SFT does not merely imitate expert policies but implicitly learns a dense, token-level reward model. The authors then recover this implicit reward from the SFT model and propose Dense-Path REINFORCE (DPR), which leverages the recovered reward to more efficiently optimize large language models (LLMs). Experimental results demonstrate that the proposed method outperforms standard SFT across multiple benchmarks.

**Strengths:**

1. The paper presents a novel idea that Supervised Fine-Tuning (SFT) can be viewed as a special case of Inverse Q-Learning, offering a new perspective for understanding SFT.

2. The authors provide a comprehensive theoretical analysis to support the proposed formulation.

3. Experimental results show considerable improvements over traditional large language model (LLM) training methods.

**Weaknesses:**

1. In the theoretical analysis, the authors make several strong assumptions about the setting—for example, assuming a deterministic token sequence and a fixed discount factor of $\gamma=1$, rather than a value smaller than 1 as typically used in RL.

2. In the proposed DPR method, the reference policy $\pi_\text{ref}$ is not formally defined. The authors state that it is an SFT checkpoint trained with half of the training samples; however, if the dataset is sufficiently large, wouldn’t this reference policy also become fully trained, thereby reducing the meaningful difference between $\pi_\text{ref}$ and $\pi_\text{SFT}$?

**Questions:**

1. The reviewer is not an expert in LLMs, but I question whether the current training paradigm of LLMs is primarily driven by SFT. Wouldn’t RLHF have a greater overall impact on model alignment and performance? I therefore have some concerns about the potential contribution and significance of this work to the broader LLM literature.

2. If SFT is theoretically equivalent to IQL, would it be possible to directly apply IQL methods to learn the reward function instead of recovering the reward from SFT?

3. The authors choose REINFORCE for policy optimization. Could the authors clarify why this choice was made instead of using more advanced RL algorithms, such as actor–critic or PPO-based methods?

---

> ### Author Response · Authors · 2025-11-24
> **Response to Reviewer HC6d [Part 1/2]**
>
> We thank the reviewer for the constructive feedback and for highlighting the novelty of framing SFT as a special case of Inverse Q-Learning. We appreciate the acknowledgment of our theoretical analysis and experimental gains. Below, we address the concerns regarding assumptions, definitions, and method choices.
> ##  Response to Weakness 1: Clarification on Theoretical Assumptions
> We appreciate the opportunity to clarify the assumptions regarding the MDP setting.
> 1. Deterministic State Transitions:
> There might be a slight misunderstanding regarding "deterministic." We do not assume the sequence generation is deterministic (the policy $\pi$ is stochastic). Instead, we assume the state transition dynamics are deterministic.
>     - In the context of autoregressive LLMs, the state $s_t$ consists of the prompt and the history of generated tokens.
>     - When an action $a_t$ (a new token) is taken, the next state $s_{t+1}$ is simply the concatenation of $s_t$ and $a_t$.
>     - Therefore, the transition function $f(s, a) = s \circ a$ is indeed deterministic by definition in standard LLM generation.
> 2. Undiscounted Rewards ($\gamma=1$):
> While $\gamma < 1$ is standard in infinite-horizon RL to ensure convergence, setting $\gamma=1$ is standard practice in the LLM post-training literature (e.g., in many RLHF implementations[link](https://github.com/OpenRLHF/OpenRLHF) and theoretical analyses[link](https://arxiv.org/pdf/2404.12358)) for two key reasons:
>     - **Finite Horizon:** Text generation is a finite-horizon task (bounded by context window or EOS). Thus, the cumulative reward is naturally bounded, and divergence is not an issue.
>     - **Long-term Dependency:** We aim for the model to weigh later tokens equally to earlier ones to ensure overall coherence. A discount factor $\gamma < 1$ would exponentially down-weight the contribution of later tokens, potentially degrading the quality of long-form generation.
> ##  Response to Weakness 2: Definition and Selection of $\pi_{ref}$
> We agree that defining $\pi_{ref}$ strictly as "50% of the data" is a heuristic derived from our specific experimental setting (100k samples), and it requires a more principled definition for generalizability.
> 1. Principle of "Incremental Competence":
> The theoretical role of $\pi_{ref}$ in our framework (Eq. 5) is to serve as a baseline that subtracts common syntactic priors and length biases. This isolates the incremental instruction-following capability gained by $\pi_{SFT}$. As demonstrated in Figure 3, performance follows an inverted U-shape relative to the baseline's training progress.
>     - **Too early:** The baseline fails to cancel out length bias and high-frequency token priors, leading to high variance.
>     - **Too late:** If $\pi_{ref} \approx \pi_{SFT}$, the reward signal (log-ratio) vanishes.
> 2. Generalization to Large Datasets:
> In our experiments, the 50% checkpoint represented the "inflection point"—where the model had acquired basic patterns of demonstration data but had not yet converged to the expert distribution. For significantly larger datasets where convergence might happen earlier (e.g., at 10% of data), we would not adhere to the 50% rule. Instead, we would select an intermediate checkpoint (e.g., based on validation loss or validation set metrics) that represents this "competent but not converged" state.

---

> ### Author Response · Authors · 2025-11-24
> **Response to Reviewer HC6d [Part 2/2]**
>
> ##  Response to Questions
>
> **Q1: Why not use RLHF?**
>
> While RLHF is a powerful paradigm, it operates under a different assumption: the availability of preference pairs or verifiable rewards.
> - **Data Accessibility: Our work focuses on the **Learning from Demonstrations (LfD)** setting, where we only have access to expert demonstrations (SFT data).
> - **Domain Adaptation:** In many practical scenarios (e.g., specialized medical or legal adaptation), collecting human preference data or defining ground-truth verification rules is expensive or infeasible.
> - **Complementary Nature:** Our method extracts denser reward signals from existing SFT data. It serves as a complementary approach when preference data is unavailable.
>
> **Q2: Can we apply IQL methods directly instead of recovering rewards from SFT?**
>
> Yes, theoretically, one could apply generic IQL algorithms.
> - **Equivalence:** Our theoretical analysis (Step 1) proves that under standard LLM assumptions (deterministic transitions, $\gamma=1$), maximizing the SFT objective is equivalent to the reduced objective of IQ-Learn.
> - **Empirical Verification:** In our early experiments, we compared a direct IQL implementation against our SFT-based reward recovery. We found no significant difference in downstream performance, which aligns with our theoretical equivalence proof.
> - **Simplicity:** Given the equivalence, we adhere to Occam's Razor by choosing the most direct method, that is SFT.
>
> **Q3: Why choose REINFORCE over Actor-Critic/PPO?**
>
> We selected REINFORCE for two primary reasons tailored to the LLM setting:
> - **Stability of Reward Estimation:** In our framework, explicitly fitting a separate critic model to approximate the undiscounted returns in long sequences often introduces approximation error and instability since undiscounted returns often have high variance. REINFORCE, using the unbiased Monte-Carlo return with our specific $\pi_{ref}$ baseline, empirically provides stable gradients without the complexity of training a Critic.
> - **Efficiency:** Recent trends in LLM post-training (e.g., REINFORCE++, GRPO) suggest that critic-free methods often achieve competitive performance with significantly lower memory and compute overhead. Since our derived reward is already dense and shaped (removing the need for a critic to handle sparse rewards), the additional cost of a critic is not necessary.

---

### Public Comment · ~Yi-Chen_Li1 · 2025-11-16
**Concerns Regarding the Characterization and Novelty Claims in Relation to Prior Work (Li et al., 2025)**

Dear Authors and Reviewers,

We are writing as the authors of "Li et al. (2025)" [arxiv link](https://arxiv.org/abs/2506.23235) to address several critical inaccuracies in how our work is presented and contextualized in this manuscript. While we appreciate the citation, the current description fundamentally misrepresents our contributions and, as a result, inflates the novelty of the present submission.

We wish to correct the record on three major points:

**1. Mischaracterization of Our Reward Formulation:**

The manuscript claims that our work's "focus is to extract sentence-level rewards." This is factually incorrect. Our paper explicitly introduces and develops a **token-level dense reward**. The core formulation presented in this manuscript is not merely similar to ours; it is identical. Specifically:
*   **Equation (4)** in this manuscript is a direct replication of **Equation (11)** in our paper (Li et al., 2025).
*   The subsequent Monte-Carlo estimation of returns described below the Equation (4) is precisely what we formulated in our **Equation (12)**.

The claim that our work is limited to "sentence-level rewards" while this submission develops a "shaping- and baseline-based reward construction" is a severe misrepresentation. Our work already established this token-level dense reward mechanism.

**2. The Claimed "Main Contribution" is Prior Work:**

The manuscript highlights the "SFT ≡ IQ-Learn equivalence at the token level" as a main contribution. This equivalence is the central theoretical pillar of our paper. We formally stated, proved, and discussed this exact concept in **Proposition 1** of our work. Therefore, this is not a novel finding of this manuscript but a core result that was previously established by Li et al. (2025).

**3. Inappropriate Use of "Concurrent Work":**

Finally, we must contest the labeling of our paper as a "concurrent effort." Our work was made publicly available on arXiv on **June 29, 2025**. The ICLR 2026 submission deadline was **September 24, 2025**. The nearly three-month gap between our public disclosure and this submission deadline provides more than sufficient time for our paper to be considered **prior art**, not concurrent work. The term "concurrent" is typically reserved for works developed and released in a much narrower, overlapping timeframe, which is not the case here. This classification minimizes the foundational nature of our contribution to the ideas presented in this submission.

In summary, this manuscript inaccurately describes our technical contributions, claims novelty for a core theoretical result we had previously published, and improperly designates our work as "concurrent."

We strongly urge the authors to issue a significant revision of the paper to accurately represent our work and properly acknowledge its contributions. We also respectfully ask the reviewers to take these points into careful consideration when evaluating the novelty and intellectual merit of this submission.

Sincerely,

Authors of (Li et al., 2025)

---

> ### Author Response · Authors · 2025-11-18
> **Response to “Concerns Regarding the Characterization and Novelty Claims in Relation to Prior Work (Li et al., 2025)”**
>
> Dear the authors of Li et al. (2025),
>
> Thank you for the detailed comments and for engaging thoughtfully with our work. We appreciate the efforts in clarifying the technical scope of the paper and recognise the importance of your contribution to the theoretical connections between SFT and IRL.
>
> We would like to respectfully address the points raised and highlight the distinction between our work and Li et al. 2025
>
>
> **1) On the SFT ≡ IQ-Learn/IRL equivalence (independent discovery)**
>
> We **independently** established the token-level equivalence between Supervised Fine-Tuning (SFT) and Inverse Q-Learning / Inverse Reinforcement Learning (IRL) before the appearance of Li et al. (2025), as recorded in our internal dated research log (June 18, 2025). Nonetheless, we agree that the paper represents a valuable and earlier public articulation of this theoretical connection, and have cited the paper in our submission
>
> Our work goes beyond this theoretical connection by developing a training algorithm with a relative baseline to stabilise the optimisation. This algorithmic contribution is distinct from and complementary to Li et al. 2025’s focus on the theoretical formulation itself.
>
> ---
>
> 2) Clarifications on equations and reward formulation
>
> **a) Mischaracterization of Reward Formulation**
>
> We appreciate the clarification that Li et al. (2025) indeed derive a *token-level dense reward*:
> \\[
> r(s_t,a_t)=Q(s_t,a_t)-\\alpha\\log\\sum_{a'}e^{Q(s_{t+1},a')/\\alpha},
> \\]
> and also present the shaped form:
> \\[
> r(s_t,a_t)=\\alpha\\log\\pi(a_t|s_t)+V(s_t)-V(s_{t+1}).
> \\]
> Their experiments, however, aggregate these rewards to form *sentence-level* scores. We acknowledge that our original description of their work ("focus is sentence-level rewards") was inaccurate. Our phrase “focus is sentence-level rewards” was meant to describe their evaluation granularity, not to deny the token-level formulation. We will revise the wording to avoid ambiguity.
>
> **b) Why our Eq. (4) and Monte-Carlo estimator match**
> The similarity between our Equation (4) and Li et al. 2025 arises naturally from the shared theoretical premise. Once the SFT ≡ IQ-Learn equivalence is proven for \\(\\gamma=1\\), the following equalities are direct consequences of the soft Bellman relation:
> \\[
> \\log\\pi(a_t|s_t)=Q(s_t,a_t)-V(s_t),\\qquad
> r(s_t,a_t)=Q(s_t,a_t)-V(s_{t+1})=\\log\\pi(a_t|s_t)+V(s_t)-V(s_{t+1}).
> \\]
> Summing across a trajectory gives
> \\[
> \\sum_{t=1}^H r(s_t,a_t)=\\sum_{t=1}^H \\log\\pi(a_t|s_t)+V(s_1)-V(s_{H+1}),
> \\]
> So the “Monte-Carlo return” naturally equals the sequence log-likelihood up to a constant. Therefore, our Eq. (4) and the corresponding estimator are not copied from Li et al. (2025); they are algebraic consequences of the shared premise. As evidence, before Li et al. (2025), Rafailov et al. (2024) [link](https://arxiv.org/abs/2404.12358) employed almost identical derivation procedures on DPO (see page 5 and Eq. (10)).
>
> **c) Algorithmic differences**
>
> We initially used **PPO** with the dense token-level reward \\(r_t=Q(s_t,a_t)-V(s_{t+1})\\). Because critic estimation is unstable when \\(\\gamma=1\\), we later switched to **REINFORCE** with a Monte-Carlo baseline and employed potential-based shaping to remove the noisy \\(V\\)-term:
> \\[
> r_t=\\log\\pi(a_t|s_t)-\\log\\pi_{\\text{ref}}(a_t|s_t),
> \\]
> which is equivalent up to a constant shift and reduces variance. These design choices—REINFORCE under \\(\\gamma=1\\), shaping-based variance control, and baseline selection—constitute core algorithmic novelties distinct from Li et al. (2025).

---

> > ### Author Response · Authors · 2025-11-18
> > **Response to “Concerns Regarding the Characterization and Novelty Claims in Relation to Prior Work (Li et al., 2025)” (Continued)**
> >
> > 3) On “Concurrent Work” and ICLR Policy:
> >
> > We used “concurrent” to denote independent development (our logs: June 18) predating their public arXiv post (June 29).
> >
> > Per ICLR 2026 policy, “arXiv is not considered a peer-reviewed venue,” [link](https://iclr.cc/Conferences/2026/ReviewerGuide#:~:text=FAQ%20for%20Reviewers), and authors are **“not required to compare to papers solely on arXiv.”** Our original citation was in good faith. We recognise that Li et al. 2025 appeared publicly earlier and will revise the phrasing to: “Independent and contemporaneous (posted earlier) work by Li et al. (2025) also explores...”
> >
> > ---
> >
> > We thank Li et al. 2025 again for their engagement. While both studies explore a shared theoretical framework, our contribution lies in extending that theory into a stable training method using a relative baseline and reward shaping. We appreciate the opportunity to clarify these points and will update our manuscript to ensure that your contributions are represented accurately and respectfully, including:
> > 1. Correct the description of Li et al. (2025) to explicitly note their **token-level dense reward** and **sentence-level aggregation**.
> > 2. Clarify that the identical forms of our Eq. (4) and Monte-Carlo return are **derivations mandated by the inverse soft Bellman operator**.
> > 3. Emphasize our independent derivation and the distinct **training recipe** under \\(\\gamma=1\\) with REINFORCE and potential-based shaping.
> > 4. Replace “concurrent” with “independent and contemporaneous (posted earlier).”
> >
> > ---
> >
> > Supplementary Material:
> > ([June 18 Work Log Recording](https://drive.google.com/file/d/1ob8hwgx9QGoe4Uah8CSOWzqzL7puv0v8/view?usp=drive_link))

---

> > > ### Public Comment · ~Yi-Chen_Li1 · 2025-11-25
> > > **Concerns Remain**
> > >
> > > Dear Authors and Reviewers,
> > >
> > > We thank the authors for their response. However, the proposed revisions are superficial and fail to address the fundamental misrepresentation of this manuscript's core contribution in relation to our prior work. The issue goes far beyond a simple choice of wording.
> > >
> > > We must insist on two points that remain unaddressed:
> > >
> > > **1. The Manuscript's Central Claim of Novelty is False.**
> > >
> > > The authors' proposed solution is to revise the text while leaving the core structure and claims of the paper intact. This is unacceptable. The **Abstract** of the manuscript, which we quote here, still presents the SFT-IRL equivalence as its primary discovery:
> > >
> > > > "In this work, we challenge this view by establishing a fundamental equivalence between SFT and Inverse Reinforcement Learning. We prove that the SFT objective is a special case of Inverse Q-Learning... This work reframes SFT not merely as policy imitation but as a powerful reward learning mechanism..."
> > >
> > > This abstract frames the "SFT ≡ Inverse RL equivalence" as the central, novel insight of *this* work. This is a direct claim to a finding that our paper (Li et al., 2025) publicly established and detailed three months prior to the ICLR submission deadline.
> > >
> > > In their own public comment, the authors now concede that their primary contribution lies elsewhere: "our contribution lies in extending that theory into a stable training method using a relative baseline and reward shaping." **This is a direct admission that the foundational theory is not their contribution.** The abstract and introduction must be fundamentally rewritten to reflect this reality, positioning the work as building upon our established theory, not discovering it. To do otherwise is to mislead the reader and the reviewers about the paper's core contribution.
> > >
> > > The claim of an "independent discovery" based on an "internal dated research log" is irrelevant to the public scientific record. **Scientific precedent is established by public disclosure, not by private notes.** Our work was publicly available and constitutes prior art.
> > >
> > > **2. The Interpretation and Application of ICLR Policy is Incorrect and Concerning.**
> > >
> > > The authors' defense regarding the "concurrent work" label is based on a misreading of ICLR's policy. The policy states:
> > >
> > > > "We consider papers contemporaneous if they are published within the last **two months**... since our full paper deadline is September 24, if a paper was published... on or after **July 24, 2025**, authors are not required to compare their own work to that paper."
> > >
> > > Our paper was uploaded to arXiv on **June 29, 2025**, which is *before* this two-month window. Therefore, it does not qualify as "contemporaneous work" but as **established prior art** under the very policy the authors cite.
> > >
> > > Furthermore, the policy is designed to protect authors who may be *unaware* of very recent work. It explicitly states: "authors are strongly encouraged to cite such related work if they are aware of it." Given that the authors have built their work on the exact same theoretical foundation and derived identical equations, their awareness of our work is not in question.
> > >
> > > The continued insistence on framing our established theoretical result as a novel contribution of your own—despite clear evidence to the contrary—is deeply troubling. The discovery of the SFT≡IRL equivalence is not a trivial "shared premise"; it is the non-obvious, foundational insight upon which both papers are built. To present this insight as your own, particularly when the key equations are identical, forces us to question whether this submission adheres to the standards of academic integrity. The similarity in core ideas and equations goes beyond coincidence and raises serious concerns about plagiarism.
> > >
> > > **Conclusion and Required Actions:**
> > >
> > > A simple rewording of the "concurrent work" section is grossly insufficient. To resolve this, a fundamental revision of the manuscript is required, starting with the **Abstract** and **Introduction**. The paper must be reframed to correctly position itself as an *application* or *extension* of the theoretical framework established in Li et al. (2025), not as its originator.
> > >
> > > Without such a change, this manuscript makes a misleading and unsupported claim to a major discovery that is not its own. We kindly ask the reviewers and the Area Chair to consider the severity of these issues when assessing the paper's novelty, contribution, and adherence to scholarly ethics.
> > >
> > > Sincerely,
> > >
> > > Authors of Li et al., 2025.

---

### Author Response · Authors · 2025-11-24
**General Response to Reviewers**

We sincerely thank all reviewers for their time and constructive feedback. We have addressed the specific questions and concerns in our individual responses to each reviewer.

**Summary of Revisions:** We would like to explicitly draw your attention to the fact that we have uploaded a revised PDF.

- **Discussion on Li et al.:** As suggested, we have updated and expanded the discussion regarding the work by Li et al. to better clarify the distinctions and contributions.

- **Location & Highlighting:** This discussion has been placed in a prominent position as the first part of Section 2 (Related Work) and is highlighted in blue for easy identification.

We hope these revisions and our responses satisfactorily address your comments. We remain open to any further questions during the discussion period.

Li et al. "Generalist Reward Models: Found Inside Large Language Models." In ArXiv 2025

---

### Meta-Review · Area_Chair_RCfu · 2026-01-11

**Summary:**

I have reviewed all author responses and find that most of the raised concerns have been adequately addressed, with only some minor issues remaining. At this point, the primary outstanding concern relates to the evaluation:

All evaluations rely on LLMs as judges. While this is an important and common metric, it is insufficient on its own to thoroughly validate the proposed ideas in the current stage of LLM development. Also as several reviewers noted, the reported gains over the baselines are relatively small. In addition, the proposed methods are evaluated on a limited set of datasets, although these datasets are widely used in prior work addressing similar problems.

Given the vital role of empirical evaluation in assessing algorithmic contributions, I believe these limitations may prevent reviewers from substantially increasing their scores. Based on this assessment, it is possible that the last reviewer may raise their score from 2 to 4; however, even in that case, the paper would likely not rise to the top tier among the submissions.

I have also carefully examined the concerns regarding potential plagiarism with respect to the related arXiv paper. Based solely on a comparison of the two manuscripts, I do not find sufficient grounds to support a plagiarism claim. Furthermore, as the other paper has not been published in a peer-reviewed venue, it should not affect the evaluation of this submission. If the arXiv authors wish to pursue a plagiarism allegation, it is their duty to provide clear and concrete evidence to substantiate the claim.

**Reviewer Concerns:**

Detailed in my meta review.

**Reviewer Scores:**

The 3rd reviewer may raise their score from 2 to 4. The others should not in my opinion.

---

### Decision · Program_Chairs · 2026-01-26

Reject